



# The determination of highly time resolved and source separated black carbon emission rates using radon as a tracer of atmospheric dynamics

Asta Gregorič[1,2], Luka Drinovec[1,3], Irena Ježek[1], Janja Vaupotič[4], Matevž Lenarčič[5], Domen Grauf[5], Longlong Wang[2], Maruška Mole[2,6], Samo Stanič[2], Griša Močnik[1,2,3]

[1]Aerosol d.o.o., Ljubljana, SI-1000, Slovenia
[2]Centre for Atmospheric Research, University of Nova Gorica, Nova Gorica, SI-5000, Slovenia
[3]Department of Condensed Matter Physics, Jozef Stefan Institute, Ljubljana, SI-1000, Slovenia
[4]Department of Environmental Sciences, Jozef Stefan Institute, Ljubljana, SI-1000, Slovenia
[5]Aerovizija d.o.o., Ljubljana, SI-1000, Slovenia
[6]Quasar Science Resources S.L., Madrid, ES-28232, Spain

*Correspondence to*: Asta Gregorič (asta.gregoric@aerosol.eu)

**Abstract.** We present a new method for the determination of the source specific black carbon emission rates. The methodology was applied in two different environments: an urban location in Ljubljana and a rural one in the Vipava valley (Slovenia, Europe), which differ in pollution sources and topography. The atmospheric dynamics was quantified using the atmospheric radon ($^{222}$Rn) concentration to determine the mixing layer height for periods of thermally driven planetary boundary layer evolution. The black carbon emission rate was determined using an improved box model taking into account boundary layer depth and a horizontal advection term, describing the temporal and spatial exponential decay of black carbon concentration. The rural Vipava valley is impacted by a significantly higher contribution to black carbon concentration from biomass burning during winter (62%) in comparison to Ljubljana (31%). Results of the calculated black carbon emission rates in Ljubljana were in the range from 280 to 300 µg m$^{-2}$h$^{-1}$ in spring and winter, respectively. Overall black carbon emission rate in Vipava valley were only 25% lower compared to Ljubljana and were in the range from 210 to 240 µg m$^{-2}$h$^{-1}$ in spring and winter, respectively. As expected, black carbon emissions from traffic prevail in Ljubljana and account for 80% of emissions during winter; the traffic contribution in the Vipava valley was only 42%. Different daily dynamics of biomass burning and traffic emissions was responsible for higher contribution of biomass burning to measured black carbon concentration, compared to the fraction of its emission rate. Coupling the high time resolution measurements of black carbon concentration with atmospheric radon concentration measurements can provide a useful tool for direct, highly time resolved measurements of the intensity of emission sources. Source specific emission rates can be used to assess the efficiency of pollution mitigation measures over longer time periods, thereby avoiding the influence of variable meteorology.



## 1. Introduction

Black carbon (BC) is an important component of fine particulate matter in the atmosphere and its most strongly light-absorbing fraction. It is produced by incomplete combustion of various carbonaceous fuels, mainly fossil fuel and biomass. Due to its

strong absorption of shortwave solar radiation, subsequent heating of the atmosphere and rapid adjustment effects on clouds and snow, it significantly contributes to the climate forcing by aerosols (Pöschl, 2005; Bond et al., 2013; IPCC, 2013). Variety of other aerosols and precursor gases are emitted together with BC, which, after the internal mixing and aging in the atmosphere, alter BC optical properties (e.g. Cui et al., 2016; Pokhrel et al., 2017) and its atmospheric removal rate due to changes of hygroscopic properties of BC-containing particles (e.g. Zhang et al., 2015). In addition, BC is an important air

pollutant decreasing local air quality and is associated with undesirable health outcomes (Janssen et al., 2011; WHO, 2012). Based on bottom-up constructed emission inventories, BC emissions from energy-related combustion have been increasing gradually from the beginning of industrial era from about 1 Tg in 1850 to 4.4 Tg in 2000 (Bond et al., 2007) and were dominated by contributions from different fuel types during technological evolution. Contribution from coal dominated until 1975, followed by biofuel and diesel-fuelled engines more recently. Considering also open biomass burning, which accounts

for 40 % of global BC emission, the total emissions in 2000 are estimated at about 7.5 Tg (Bond et al., 2013). According to the study of Klimont et al. (2017), where additional emission sources were considered, global BC anthropogenic emissions were estimated at about 6.6 and 7.2 Tg in 2000 and 2010, respectively, with a contribution of 0.7 Tg from Europe and Russia in 2010. Reduction of BC emission factors by implementation of cleaner technology partly offset the rapidly increasing energy consumption since 1950, resulting in slower growth of particulate matter emissions in comparison to $CO_2$. Although current

emission inventories agree quite well on the main emission sources and regions, there exist significant uncertainties in the emission factors and activity data, used for emission calculation, with recent observationally constrained estimations much higher than the ones traditionally used (Sun et al., 2019). In contrast to bottom-up emission inventories, top-down constrained methods (such as inverse modelling) focus at minimising the difference between simulated pollutant concentration, based on estimated emission flux, and measured pollutant concentration (Brioude et al., 2013; Wang et al., 2016b; Guerrette and Henze,

2017). These methods can provide spatially and temporally better resolved assessment of pollutant emissions, including BC, but are influenced by different sources of uncertainty, mainly from the insufficient evaluation of long-range transport of polluted air-masses.

According to the European Union emission inventory report (LRTAP, 2018), 0.2 Tg of BC were emitted in 2016 in the EU-28 region, with the dominant energy-related emissions from on-road and non-road diesel engines accounting for about 70 %

of all anthropogenic BC emissions (Bond et al., 2013). Recently updated United States black carbon emission inventory (Sun et al., 2019) pointed out decreasing trend of BC emissions from 1960 to 2000, dominated by vehicle, industrial and residential sectors. Traffic related BC emission is dominating primary particulate matter (PM) emission especially in major cities (e.g. Pakkanen et al., 2000; Klimont et al., 2017). Recently biomass combustion for residential heating has been promoted under





the label of renewable fuel and additionally increased due to economic crises and increase of other fuel prices (Crilley et al., 2015; Denier van der Gon et al., 2015; Hovorka et al., 2015; Athanasopoulou et al., 2017). Although several studies report significant role of wood burning emissions on BC concentrations in Alpine valleys (Sandradewi et al., 2008b; Favez et al., 2010; Herich et al., 2014) and Scandinavian rural areas (Ricard et al., 2002; Aurela et al., 2011), increase in contribution of

wood smoke to fine PM was noticed also in several large urban areas (Favez et al., 2009; Crippa et al., 2013; Fuller et al., 2014; Denier van der Gon et al., 2015; Hovorka et al., 2015; Helin et al., 2018; Zhang et al., 2019). Notable contribution of wood smoke was observed also in Slovenian urban (Ogrin et al., 2016) and rural areas (Wang et al., 2019), responsible for air quality deterioration especially in geographically constrained areas such as valleys and basins.

Since BC is a chemically inert primary pollutant, it is a good measured indicator of emissions and can provide valuable

information to authorities in the implementation and evaluation of air quality action plans, by indicating the strength of different emissions sources (e.g. Reche et al., 2011; Titos et al., 2015). On the other hand, emission inventories provide an important information for climate models by providing data about the changing pattern of BC emissions, its major sources and historical evolution. From the perspective of short-term local air quality prediction, improving local diurnal and seasonal pattern of BC emissions would greatly benefit the model prediction performance. Atmospheric chemical transport models based on the

fundamental description of atmospheric physical processes can improve the knowledge about temporal evolution of BC emissions at the modelled area, but require comprehensive input data of atmospheric processes (Seinfeld and Pandis, 2016).

Besides the intensity of pollutant emission, micrometeorology of the planetary boundary layer (PBL) plays an important role in controlling time evolution of pollutant concentration. PBL is the lowest part of the troposphere, which is directly affected by the presence of the Earth's surface. Soon after sunrise, convectively driven heat transfer from the ground surface and

mechanical mixing (due to wind shear and surface roughness) is responsible for the formation of the turbulent mixing layer (ML), which grows by entraining the air from above and reaches its maximum depth in the late afternoon. Thus, the mixing layer height (MLH) strongly depends on the intensity of solar radiation reaching the ground. Entrainment of air from above takes place in a stable layer at the top of the ML, the so-called entrainment zone. The residual layer is formed after the decay of turbulence shortly before sunset, with its bottom portion transformed into a stable nocturnal boundary layer (SNBL) during

the night. SNBL is characterized by stable stratification with low mixing. The bottom 10 % of the boundary layer, either ML or SNBL, represents the surface layer, where turbulent fluxes and stress vary by less than 10 % of their magnitude (Stull, 1988). Anthropogenic air pollutants are generally emitted from the surface and trapped within the PBL, where their concentrations are controlled by the turbulent mixing (e.g. Quan et al., 2013; McGrath-Spangler et al., 2015; Tang et al., 2016). Therefore, on diurnal timescales, atmospheric stability/dynamics (tendency of the atmosphere to resist or enhance the initial

displacement of air parcels) plays a key role on the variability of primary inert pollutants, such as BC, and is affected by them (e.g. Ferrero et al., 2014).

Different approaches for MLH determination are reflected in different definitions found in literature (Seibert et al., 2000). One of generally adopted definitions relies on the vertical extent of dispersion of pollutants, which are released from the surface within 1-hour time period. However, since slope winds may influence vertical pollutant transport, this definition may be



ambiguous at complex terrains such as Alpine valleys (Leukauf et al., 2016). MLH can be determined by indirect methods based on remote sensing using lidars (Caicedo et al., 2016; de Bruine et al., 2017) or radiosounding measurements, by applying different measures, such as Richardson number, Monin-Obukhov length or turbulence kinetic energy (Stull, 1988). These methods have usually low spatial and/or temporal resolution. Furthermore, interpretation of mixing in very stable conditions

is challenging due to phenomena such as nocturnal jets and local flow perturbations (Williams et al., 2013), leading to non-reliable representation of stable atmospheric conditions by numerical weather or dispersion models.

Assessment of the BC emission rate requires decoupling of meteorologically driven variation from the dynamics of the sources. An alternative way to overcome the difficulty associated with the proper physical interpretation of micrometeorological properties of the ML and dispersion characteristics is the use of a tracer method. Naturally occurring noble radioactive gas

radon ($^{222}$Rn) has been applied in the past for the study of long-range transport of air masses (Hansen et al., 1990; Crawford et al., 2007) and recently for studying PBL characteristics (e.g. Griffiths et al., 2013; Williams et al., 2013; Pal et al., 2015; Salzano et al., 2016; Vecchi et al., 2018), microclimate spatial variability (Chambers et al., 2016; Podstawczyńska, 2016) and impact assessment of atmospheric stability on local air pollution (Perrino et al., 2001; Chambers et al., 2015a; Chambers et al., 2015b; Crawford et al., 2016; Wang et al., 2016a; Williams et al., 2016). Good correlation, at least for the periods of thermally

driven PBL convection, was observed in previous studies comparing effective MLH derived by the box model and MLH obtained by modelling approaches based on turbulence variables (Allegrini et al., 1994; Vecchi et al., 2018; Kikaj et al., 2019) or remote sensing techniques (sodar, lidar) (e.g. Salzano et al., 2016). Kikaj et al. (2019) successfully identified persistent inversion events in the Ljubljana basin based on $^{222}$Rn measurements. The emission pattern of gaseous traffic related air pollutants in Bern (Switzerland), estimated by a box-model based on radon tracer, with an included advection term, showed

an excellent correlation with traffic density (Williams et al., 2016). These studies imply that radioactive tracer method gives reliable information on the effective mixing layer height and indication of atmospheric stability (e.g. Perrino et al., 2001), which can be easily implemented in the environmental monitoring networks.

Radon is the only gaseous element in the $^{238}$U radioactive decay chain, with its only significant sources in the natural environment being rocks and soil. After emanation from rock surfaces and soil grains, it is transported by diffusion and

advection to the surface by carrier gasses (Etiope and Martinelli, 2002). Once exhaled from the Earth's surface, it is subjected to the mixing within the ML, thus experiencing the same extent of dispersion as other air pollutants emitted from ground-based sources. As a noble gas with a relatively short half-life (3.82 days), it represents an ideal tracer for the study of PBL processes. The strength of radon flux from the surface to the atmosphere, the so-called radon exhalation rate ($E_{Rn}$), depends mainly on the surface permeability and the radon potential (Karstens et al., 2015). Surface permeability is controlled by geological setting

– lithological (rock type) (e.g. Kardos et al., 2015), structural characteristics (presence of fault zones) (e.g. Vaupotič et al., 2010) and soil moisture. Fault zones can act as a pathway for different gaseous species especially in the geologically active environments, leading to anomalies in $E_{Rn}$, whereas $E_{Rn}$ above karst caves and fractured rocks can experience high spatial and temporal variability (Vaupotič et al., 2010; Gregorič et al., 2014). Since radon is poorly soluble in water, its only sink is the radioactive decay, which can be neglected on hourly bases. As reported by Vaupotič et al. (2007) measured $E_{Rn}$ on Slovenian





territory spans over large range from few tens to several hundred mBq m$^{-2}$ s$^{-1}$. Radon exhalation rate is usually considered constant on short temporal scales in areas with homogeneous geologic characteristics (Pearson and Jones, 1965). However, exhalation of Rn is a complex process which can be assessed with different modelling approaches. Salzano et al. (2016) showed, that the error in the modelled effective MLH by considering constant radon source can be up to 10 %. Local

heterogeneity of $E_{Rn}$ due to heterogeneous soil permeability (within few meters range) is homogenised in the thin atmospheric layer (~0.5 m) close to the ground and does not represent a significant concern for measurements above this height. On a seasonal scale, however, $E_{Rn}$ decreases with the presence of snow cover, frozen soil or during and shortly after (on time scale of few days) rainy periods, due to reduced surface permeability, thus representing one of the main sources of uncertainty in the box model. It is also worth noting, that reliable exhalation measurements (used for the box model) should be conducted in

a broad network in the extent of modelled area in different periods of year, so the seasonal changes in soil permeability would also be considered. The seasonal changes of $E_{Rn}$ were already pointed out in different studies (Salzano et al., 2016; Williams et al., 2016; Chambers et al., 2019).

The aim of this paper is to determine the BC emission rate apportioned to traffic and biomass burning sources, its diurnal pattern and monthly variation for two distinct locations in Slovenia (Europe), which differ from the point of view of their

natural characteristics (geology, geomorphology, meteorology) and urban environment (urban and rural background). Both sites are subjected to their own pattern of air pollution episodes which will be addressed and interpreted based on Eulerian box model. The effective mixing layer height will be reproduced for both sites based on Rn measurements, taking into account seasonally resolved $E_{Rn}$, then used for decoupling meteorologically driven changes of measured BC concentration from the one resulting from the source dynamics. The highly time-resolved and source apportioned BC emission rate ($E_{BC}$) represents

an essential information for short-term forecasts of air pollution episodes, as well as for the evaluation of the efficiency of air quality abatement measures and their potential adaptation. Temporal variation of BC concentration will be highlighted from the point of view of PBL evolution.

A list of acronyms and symbols used in this paper is given in Table 1.
.





## 2. Method

### 2.1 Measurement locations

Two distinct measurement locations were selected for this study. The first one is located in the urban area of Ljubljana (LJ, capital city of Slovenia), which lies in the central part of Slovenia (Europe) (Figure 1). The second measurement location was

in a small town Ajdovščina (AJ), located in the Vipava valley in the western part of Slovenia. Measurement campaigns lasted from autumn 2016 to spring 2017 (November 23, 2016 to May 20, 2017) in Vipava valley and from winter 2017 to summer 2017 (February 1, 2017 to June 8, 2017) in Ljubljana. Due to its basin location, Ljubljana is characterised by poor ventilation and frequent occurrence of temperature inversions, which constrains pollutants emitted from surface sources within the limited air volume inside the basin. During stable atmospheric conditions, especially in the SNBL, a thin layer of drainage winds

(colder air, adjacent to the ground, flowing downhill under the influence of gravity) and a flow of air from the edge of the city towards the centre (formed due to the heat island) governs air circulation within the Ljubljana basin (Stull, 1988; Ogrin et al., 2016). During the measurement campaign, around 10 cm of snow cover was present from January 14 to February 2, 2017.

The population of the wider Ljubljana basin is around 500,000, of which 287,000 live in the urban municipality of Ljubljana. Much more than 100,000 daily commuters from other municipalities represent additional traffic flow on working days. (Ogrin

et al., 2016). Rapid growth of automobile use is observed in the last few decades, leading to daily traffic jams inside the city and its surroundings. In the recent years, the local municipality has implemented different measures of sustainable mobility in order to improve the air quality. In particular, the traffic restriction in the major road in the city centre has led to 70% decrease of local BC concentrations (Titos et al., 2015). Besides traffic-related air pollution, emissions from combustion of biomass fuel for residential heating are a significant source of particulate concentrations in the whole country (Gjerek et al.,

2018), not only in rural areas, but in Ljubljana as well. Although district heating is provided in several areas of the city, the use of wood boilers and fireplaces is a common practice.

The population of the second area of interest, including surrounding villages is relatively small, with about 19,000 residents living in the municipality of Ajdovščina, of which about 7,000 residents live in the town. The Vipava valley is confined to the north by the steep ridge, which rises up to about 1000 m a.s.l., and to the south by the Karst plateau with an average altitude

of about 300 m. Due to the complex topography, the valley usually experiences two extreme cases of atmospheric stability conditions. On one side, stable atmospheric conditions can last for several days, leading to the formation of strong vertical aerosol gradients, which are followed by frequent occurrences of strong downslope Bora wind (Mole et al., 2017; Wang et al., 2019). A highway connecting the central part of Slovenia with Italy runs through the valley on the southern border of the town, around 800 m away from our measurement site. The Mediterranean climate of this area is responsible for mild winters and

warm summer season, with residential heating mainly limited to the cold season, from November to February, with biomass fuel being the primary source of energy.



**Geological setting**

From geological point of view, the city of Ljubljana lies in the neotectonic basin with extensive and thick accumulations of Quaternary glaciofluvial sediments on the northern and central parts (gravel and conglomerate), whereas the south-western part of Ljubljana basin is filled with lacustrine and paludal sediments (Janža et al., 2017). The maximum thickness of sediments

is around 170 m. Non-consolidated Quaternary sediments are permeable enough to allow spatially and temporally homogeneous Rn exhalation rate.

The geological structure of the broader area of Vipava valley results from Tertiary thrust of Cretaceous limestone, which forms the steep north-eastern ridge of the valley, on the Eocene flysch rocks, forming the valley floor. Flysch rocks consist of alternating layers of marlstone and carbonatic sandstone. Due to physical weathering of the limestone, a large amount of

limestone scree material has been formed and deposited on the underlying flysch rocks on the slopes of the north-eastern ridge. Valley floor is covered by clayey weathered residual of flysch rocks with fine flysch scree (Jež, 2007). Spatially homogeneous Rn exhalation rates can be expected along the valley.

## 2.2  Black carbon measurements and source apportionment

Measurements of black carbon concentration were conducted at two sites at each location, Ljubljana and Vipava valley, one at the urban background and the other one at the higher altitude (Figure 1), in order to provide an insight into the extent of the vertical aerosol mixing. Periods during which similar BC concentrations were measured at both sites indicate time periods when MLH reached or exceeded altitude difference between both sites. Although this kind of measurement composition certainly indicates periods of stable PBL conditions, detection of the exact time of MLH reaching the upper site is more

uncertain due to diffusion mixing (in the case of Ljubljana) or local slope winds (Leukauf et al., 2016) (Vipava valley) and therefore it does not provide undisturbed MLH evolution characteristic for the whole valley/basin.

The urban background site of the Slovenian Environmental Agency (ARSO) was used for BC measurements in the city of Ljubljana (295 m a.s.l.), while measurements at the Golovec Astronomical and Geophysical Observatory (GOL), 100 m above Ljubljana city (395 m a.s.l.) were used as the hill site. The inlet at ARSO was about 4 m above the ground, while the inlet at

GOL was about 2.5 m above ground. Measurements in the Vipava valley were conducted about 20 m above ground level (120 m a.s.l) on the roof of the building of University of Nova Gorica, located in the town of Ajdovščina (AJ). About 830 m higher (950 m a.s.l.), on the north-eastern ridge of the valley, the second measurement site was installed at the Otlica Meteorological observatory (OT) of the same university.

Aerosol light absorption and corresponding mass equivalent black carbon concentration (BC) was measured at 7 different

wavelengths (370 – 950 nm) using the Aethalometer model AE33 (Magee Scientific / Aerosol d.o.o.), with "dual spot" technique used for real-time loading effect compensation (Drinovec et al., 2015). Flow rate was set to 5 l/min and the measurement time resolution to 1 minute. TFE-coated glass fibre filter was used with multiple scattering parameter (C) set to





1.57. The mass absorption cross section $\sigma_{air}$ of 7.77 m$^2$ g$^{-1}$ was used to convert the optical measurement at 880 nm to BC mass concentration.

Aethalometer measurements at different wavelengths provide an insight in the chemical composition of light absorbing particles. The so called Aethalometer model (Sandradewi et al., 2008a) was used to apportion BC to traffic and biomass burning

sources. The model uses an a priori assumed pair of absorption Ångström exponents (AAE) for traffic (AAE$_{TR}$) and biomass burning (AAE$_{BB}$) to determine the contribution of both sources. A narrow range of AAE$_{TR}$ (0.8 – 1.1) values is reported in the literature, whereas larger AAE$_{BB}$ values (from about 1.5 up to 3.5) in the wider range are characteristic for biomass burning sources (Kirchstetter, 2004; Saleh et al., 2013; Garg et al., 2016; Zotter et al., 2017). Higher values of AAE$_{BB}$ result from enhanced light absorption in the near-UV and blue part of the spectrum caused by organic carbon species, present in biomass-

smoke. Source specific AAE can be independently determined using auxiliary measurements of OC/EC. and $^{14}$C (Sandradewi et al., 2008a; Zotter et al., 2017), or biomass burning tracers like levoglucosan (Favez et al., 2010; Herich et al., 2014; Hellén et al., 2017; Helin et al., 2018). Since independent measurements allowing the determination of the AAE pair representative for our measurement locations were not available, the most suitable AAE pair was estimated according to the commonly used AAE values published in the literature, by considering overall distribution of AAE (Figure S1) for each measurement location

and the corresponding diurnal variation of traffic ($BC_{TR}$) and biomass burning related BC ($BC_{BB}$) (Figure S2). AAE was calculated using the Eq. 1 for 470 nm and 950 nm wavelengths, where $b_{abs}$ stands for the absorption coefficient at 470 nm and 950 nm.

$$AAE = \frac{\ln\left(\frac{b_{abs(470)}}{b_{abs(950)}}\right)}{\ln(950/470)} \qquad (1)$$

By taking into account equations provided by Sandradewi et al. (2008a), the $BC_{BB}$ and $BC_{TR}$ were finally calculated using the Eq. 2 and 3, respectively.

$$BC_{BB} = \frac{\frac{b_{abs(470)}}{b_{abs(950)}} - \left(\frac{950}{470}\right)^{AAE_{TR}}}{\left(\frac{950}{470}\right)^{AAE_{BB}} - \left(\frac{950}{470}\right)^{AAE_{TR}}} \times BC \qquad (2)$$

$$BC_{TR} = BC - BC_{BB} \qquad (3)$$

AAE$_{TR}$ / AAE$_{BB}$ pair of 1.0 and 2.0, respectively, was chosen for both measurement locations. Further discussion on the choice of AAE pair used for source apportionment is provided in the Section 1 of the Supplement.





### 2.3 Radon measurements

Radon measurements were conducted at both measurement locations, in the city of Ljubljana on the floor of the basin and on the floor of Vipava valley, close to the town of Ajdovščina. Measurements cover longer period than BC measurements: from November 11, 2016 to May 31, 2017 in Vipava valley (with about one-month gap in March 2017 due to instrument malfunction) and from December 14, 2016 to June 8, 2017 in Ljubljana. At both measurement sites, instruments were installed outdoors under a roof of single-family house (to shelter instruments from environmental effects, but otherwise open), surrounded by unperturbed natural soil ground. Both sites were selected in the area that is subjected to the same boundary layer characteristics as aerosol measurements. Instruments were installed 1 m and 3 m above ground in Ljubljana and Vipava valley, respectively. Radon activity concentration ($C_{Rn}$) was measured using AlphaGUARD PQ2000 PRO (Bertin Instruments) radon monitor. In the instrument, the measured gas diffuses through a large-surface glass fibre filter into the ionization chamber. The instrumental lower limit of detection is 2-3 Bq m$^{-3}$ at 1-hour time resolution. In order to decrease noise, radon measurements were smoothed by applying an FFT filter with cut-off frequency of 0.25 h$^{-1}$. Comparison of smoothed and raw Rn measurements is shown in Figure S6. Due to sampling in diffusion mode, 1 hour time lag was considered when combining $C_{Rn}$ data with other measurements.

### 2.4 Meteorological parameters and supporting information

Meteorological parameters, such as air temperature ($T$), wind speed and direction ($ws$, $wd$), amount of precipitation and snow cover were provided by the Slovenian Environment Agency (ARSO) for the Ljubljana measuring location. In the Vipava valley, all meteorological parameters are measured at the meteorological station situated at the valley floor close to the town of Ajdovščina, whereas wind data were collected at the rooftop of the University of Nova Gorica building in Ajdovščina as a part of research conducted at the Centre for Atmospheric Research (Mole et al., 2017).

MLH dataset, obtained from the NOAA Air Resources Laboratory (NOAA-ARL) Global Data Assimilation System (GDAS) database (Rolph et al., 2017), was considered as supplementary information for comparison with the effective MLH values derived from the box model. Archived dataset has spatial resolution of 1° and temporal resolution of 3 hours. Although spatial resolution is not fine enough to capture local micrometeorological characteristics, it gives an estimation on the wider area PBL stability and effective mixing height.

Traffic counts data for Ljubljana were provided by the Municipality of Ljubljana for the whole period of measurements for several different locations within the city.

Two complementary methods were used for detection of MLH and comparison with MLH derived by the box model. Scanning mobile Mie-scattering lidar is used at the University of Nova Gorica for studies of Bora wind (Mole et al., 2017), aerosol properties and PBL characteristics (Wang et al., 2019) in the Vipava valley. Detailed lidar configuration and performance is provided by He et al. (2010). In our study, MLH was estimated based on the retrieval of range-corrected lidar signal in selected





periods. On the other hand, vertical BC concentration profiles were measured over the Ljubljana basin by ultralight aircraft on selected days from February to May 2017. A lighter modified version of Aethalometer AE33 with an isokinetic sampling inlet was used for BC vertical profile measurements. Measurements provided useful information about aerosol vertical dispersion characteristics and MLH estimation (Ferrero et al., 2011). Further details about analyses approach is provided in the

Supplement (Section 2).

### 2.5 Radon-based MLH modelling

The box model approach introduced in previous studies (Sesana et al., 2003; Griffiths et al., 2013; Salzano et al., 2016; Williams et al., 2016; Vecchi et al., 2018) employs the Eulerian approach, including a constant radon source ($E_{Rn}$) and vertical

entrainment of air masses from the residual layer. Salzano et al. (2016) improved the model performance by considering the variability in the soil radon exhalation rate, where the authors showed up to 10% difference in modelled MLH compared to the model using constant Rn source. In this paper we applied the approach introduced by Williams et al. (2016), where an inclusion of a simplified horizontal advection term allows the quantification of local emissions of air pollutants.

Considering a vertically well mixed box (box dimension discussed in detail in Sec. 2.6.1) with species concentration $C_s$, the

mass of species "$s$" in a column of air within the effective MLH ($h$) over one m$^2$ at time $t_i$ depends mainly on the emissions ($E_s$) from the surface and the remaining mass of species from the previous time period ($t_{i-1}$). When ML is growing ($c_s^+$), there is an additional encroachment of species, which remained in the residual layer from the previous day, while in the case when this layer is shrinking ($c_s^-$), a part of mass is considered to be removed from the mixing layer (Eq. 4).

$$C_{s_i}h_i = E_s\Delta T_s + C_{s_{i-1}}h_{i-1}e^{-\lambda_s' dt} + c_{s_{i-1}}^{\pm}(h_i - h_{i-1})e^{-\lambda_s' dt} \qquad (4)$$

The term $\Delta T_s$ (Eq. 5) includes correction due to decay of species in time period $dt = t_i - t_{i-1}$, which is characterized by decay constant $\lambda_s'$. If decay rate is very slow, decay of species within $dt$ becomes negligible, thus $\Delta T_s \rightarrow dt$.

$$\Delta T_s = \frac{1-e^{-\lambda_s' dt}}{\lambda_s'} \qquad (5)$$

Decay constant $\lambda_s'$ $[h^{-1}]$ takes into account temporal decay and horizontal advection. The temporal decay constant $\lambda_s$ $[h^{-1}]$ accounts for internal sinks due to chemical reactions, dry and wet deposition or radioactive decay. Horizontal advection assumes exponential decrease of species concentration downstream (Eq. 6) and is characterized by spatial decay constant $\gamma_s$ $[m^{-1}]$:

$$\lambda_s' = \lambda_s + U\gamma_s \qquad (6)$$

where $U$ represents layer averaged wind speed. A small uncertainty is introduced to the model, since wind data were available only from standard meteorological measurements at the height of 2 m above the surface.




Three different cases can be parametrized during a course of a day:

1. During stable conditions, when $h_i = h_{i-1}$, Eq. 4 is reduced to Eq. 7:

$$C_{s_i} - C_{s_{i-1}}e^{-\lambda'_s dt} = \frac{E\Delta T_s}{h_i}$$ (7)

2. After the sunrise when PBL starts to grow ($h_i > h_{i-1}$), a volume of air mass from the residual layer is incorporated into the expanding ML and the term $c_s^{\pm}$ from the last part of Eq. 4 is modelled as $c_s^+$ (Eq. 8):

$$c_{s_{i-1}}^+ = c_s^0 e^{-\lambda'_s(t_{i-1}-t_0)}$$ (8)

where $c_s^0$ is species concentration from the previous day at time $t_0$, just before the afternoon transition to SNBL. $c_s^+$ represents the concentration of species in the residual layer.

3. When PBL is shrinking ($h_i < h_{i-1}$) a volume of air is decoupled from ML and forms the residual layer. $c_s^{\pm}$ of Eq. 4 is set to $c_{s_{i-1}}^- = C_{s_{i-1}}$.

The first phase of modelling is focused to the quantification of the effective MLH based on atmospheric Rn concentration measurements. Rn data with 1 hour time resolution were first smoothed by applying FFT filter with cut-off frequency of 

0.25 h$^{-1}$ in order to decrease noise level, since small changes of Rn concentration can cause unexpected fluctuation in the calculated MLH. Assuming constant, spatially homogeneous radon source, which extends beyond the limits of our modelled area, spatial decay constant in Eq. 6 can be approximated to zero ($\gamma_{Rn} = 0$) and decay constant $\lambda'_{Rn}$ is equal to radon radioactive decay constant: $\lambda_{Rn} = 0.0076\ h^{-1}$. Radioactive decay accounts for less than 1% of $C_{Rn}$ decrease during the course of one hour.

For stable atmospheric conditions, when $h_i = h_{i-1}$, Eq. 4 is simplified to

$$h_i = \frac{E_{Rn}\Delta T_{Rn}}{dC_{Rn}}$$ (9)

where $dC_{Rn}$ represents difference in radon concentration measured in the time period $dt$:

$$dC_{Rn} = C_{Rn_i} - C_{Rn_{i-1}}e^{-\lambda_{Rn}dt}$$ (10)

Condition of expanding or shrinking ML is tested by comparing difference of concentration with emission of Rn to the ML with MLH = $h_{i-1}$ in the same time period. In the case of expanding ML, change of Rn concentration is smaller than expected for stable MLH: $dC_{Rn} < E_{Rn}\Delta T_{Rn}/h_{i-1}$ (Eq. 11), whereas in the case of shrinking ML, concentration increases faster than it would be expected for stable MLH: $dC_{Rn} > E_{Rn}\Delta T_{Rn}/h_{i-1}$ (Eq. 12). Effective MLH is then calculated for the two separate conditions as:

$$h_i = \frac{E_{Rn}\Delta T_{Rn} + h_{i-1}\left(C_{Rn_{i-1}} - C_{Rn_{i-1}}^+\right)e^{-\lambda_{Rn}dt}}{C_{Rn_i} - C_{Rn_{i-1}}^+ e^{-\lambda_{Rn}dt}}$$ (11)

$$h_i = \frac{E\Delta T_{Rn}}{C_{Rn_i} - C_{Rn_{i-1}}e^{-\lambda_{Rn}dt}}$$ (12)





where $C_{Rn}^{+}$ represents radon concentration remaining after decay in the residual layer from the previous afternoon.

When MLH reaches its full extent in the late afternoon, it can extend above the previous day's residual layer, thus incorporating Rn "free" air into the ML. In these conditions $C_{Rn}$ reaches its lowest daily concentration, which can be similar or even lower than concentration from the previous day's residual layer ($C_{Rn}^{+}$). In such conditions, calculation following Eq. 11 becomes unstable with high uncertainty, leading to significant overestimation of the effective MLH. Since incorporation of residual layer into the Eq. 11 resulted in most cases in non-meaningful determination of MLH in the couple of hours preceding the afternoon transition to SNBL, residual layer Rn concentration was set to zero ($C_{Rn_{i-1}}^{+} = 0$).

**Determination of radon exhalation rate - $E_{Rn}$**

The results of MLH determined by the box model strongly depend on correct estimation of radon exhalation rate. As discussed in previous sections, $E_{Rn}$ is affected by seasonal meteorological changes mostly by varying soil humidity and permeability. Since continuous monitoring of $E_{Rn}$ is usually not available, the box model has to be calibrated to any available information of MLH. In our study we used the modelled MLH for the determination of $E_{Rn}$ in the following way:

- $E_{Rn}$ was considered to be stable during the course of one month. In this regard, periods of specific microclimatic conditions, such as rainy and windy days were removed from the analyses.
- Modeled MLH was obtained from the Air Resources Laboratory (NOAA-ARL) Global Data Assimilation System (GDAS) database, noting the limited spatial and time resolution of the model.
- MLH was calculated using Eqs. 9, 11, 12 for different values of radon exhalation rates in the range from 50 to 600 Bq m$^{-2}$h$^{-1}$ (Rn-model).
- GDAS-modeled-MLH was correlated with Rn-model derived one for each choice of exhalation rate at the GDAS time resolution. Correlation slopes are presented in Figure 2. We obtained the monthly average exhalation rate at unity correlation slope.

This method allowed us to obtain average monthly exhalation rate (Table 2) using the data from the meteorological model (GDAS), even though the model spatial and time resolution is low. The uncertainty of $E_{Rn}$ is assessed from the slope error and was estimated to be 50 Bq m$^{-2}$ h$^{-1}$. The obtained exhalation rates were validated by comparing Rn-model MLH to MLH determined from vertical profiles of BC measured with an aircraft (Supplement Section 2) and show good agreement at the time of the airplane flights.





### 2.6 Black carbon emission rate modelling

The second part of modelling uses the box model (Eq. 4), where measured BC concentrations, apportioned to sources, are inverted, taking into account effective MLH determined during the first step, to calculate hourly resolved BC emission rate ($E_{BC}$).

1-minute dataset of source apportioned BC concentration was first averaged to 1-hour time base in order to correspond to the determined MLH values. BC emission rates were calculated separately for traffic ($E_{TR}$) and biomass burning emissions ($E_{BB}$), using Eq. 13 and 14, for increasing and decreasing MLH, respectively:

$$E_{BC} = \frac{1}{\Delta T_{BC}}\left(C_{BC_i}h_i - C_{BC_{i-1}}h_{i-1}e^{-\lambda'_{BC}dt} - C^+_{BC_{i-1}}(h_i - h_{i-1})e^{-\lambda'_{BC}dt}\right) \tag{13}$$

$$E_{BC} = \frac{h_i}{\Delta T_{BC}}\left(C_{BC_i} - C_{BC_{i-1}}e^{-\lambda'_{BC}dt}\right) \tag{14}$$

where the index $i$ represents the traffic ($TR$) or biomass burning ($BB$) contributions to BC concentrations, and $\lambda'_{BC}$ is the decay constant calculated by Eq. 6, which accounts for temporal decay and horizontal advection. The latter is introduced due to dispersion characteristics and inhomogeneous spatial distribution of emission sources, which usually leads to decrease of species concentration downstream. In the Eulerian box model, the difference between species concentration within the modelled area and outside the box controls the spatial decrease of the concentration downstream. Based on the study presented by Williams et al. (2016), an assumption of exponential decay was considered in this study to simplify the model and overcome the fact of missing measurements at the box outer limits. The size of the modelled area and distribution of specific sources leads to source specific spatial decay constants, namely $\gamma_{BB}$ for biomass burning and $\gamma_{TR}$ for traffic related BC. Spatial decrease of BC concentration with distance from the source for different choice of $\gamma$ is presented in Figure S9 a.

Since BC particles are inert, the rate of BC removal from the atmosphere is governed by wet deposition (e.g. Blanco-Alegre et al., 2019). The temporal sink was estimated based on mean life-time of soot particles in the atmosphere, which can be considered between 1 week and 10 days (Cape et al., 2012). Therefore $\lambda_{BC}$ of 0.006 h$^{-1}$ was considered in this study, corresponding to 1 week mean lifetime of BC. The same temporal sink was considered regardless of BC source.

Data analysis and graphical representation was performed using R programming language (R Core Team', 2018), with "ggplot2" (Wickham, 2009), "openair" (Carslaw and Ropkins, 2012) and "dplyr" packages. If not stated otherwise, time is reported as local time (CET/CEST). Seasonal statistics was computed considering December – February as winter, March – May as spring, June – August as summer and September – November as autumn.

### 2.6.1 Determination of decay constants

Horizontal advection dominates over BC temporal sink, which is responsible for a small offset in modelled emission rates. A longer estimated lifetime of BC particles would result in lower modelled emission rates. Changing $\lambda_{BC}$ from 0.006 h$^{-1}$ (mean





lifetime of 7 days) to 0.004 h$^{-1}$ (mean lifetime of 10 days) would lower the average BC emission rate by approximately 15%. On the other hand, horizontal advection as parametrized by the estimated spatial decay constant, has much stronger influence on the calculation of BC emission rates. Since horizontal advection strongly depends on wind speed, the total decay constant ($\lambda'_{BC}$) also follows diurnal wind pattern with the highest values in the afternoon (Figure S5). When $ws \rightarrow 0$, the only process

responsible for decrease of BC concentration is its temporal sink. With higher wind speed, concentration would decrease exponentially. Previous studies of BC source apportionment and distribution of BC apportioned to traffic and biomass burning sources, performed in the Ljubljana basin, have revealed a homogeneous distribution of $BC_{BB}$, while $BC_{TR}$ showed a stronger dependence on the proximity of traffic sources (Ogrin et al., 2016). Therefore, $\gamma_{BB}$ for the Ljubljana basin was selected based on the general area contributing to $BC_{BB}$ concentrations and was set to $5 \times 10^{-5}$ m$^{-1}$ (Table 3) which corresponds to half-

distance decay of approximately 14 km. On the other hand, a smaller contributing area was chosen for $BC_{TR}$, thus $\gamma_{TR}$ was set to $7 \times 10^{-5}$ m$^{-1}$ (corresponding to 10 km half-distance decay). A comparison with the traffic density shows that the most suitable $\gamma_{TR}$ for Ljubljana is found in the range from $5 \times 10^{-5}$ m$^{-1}$ to $7 \times 10^{-5}$ m$^{-1}$ (Figure S9 b). Overestimating horizontal advection (an overestimation of the $\gamma$ value) would result in an overestimation of $E_{TR}$ (and $E_{BB}$), which would be especially pronounced during the periods of stronger wind speed, thus in the afternoon, which would result in an altered $E_{TR}$

diurnal pattern. Results of sensitivity analyses of modelled $E_{TR}$ for different $\gamma_{TR}$ values, which was performed based on comparison with measured traffic density at representative location close to BC measurement site in Ljubljana, is presented in the Supplement (Section 5).

Vipava valley is geographically constrained to smaller area, with small urban centre, widespread distribution of individual houses and highway running along the valley. Considering these characteristics, $\gamma_{BB}$ and $\gamma_{TR}$ were both set to $10^{-4}$ m$^{-1}$ (7

km half-distance decay).

### 3. Results

#### 3.1 Radon concentration and meteorological conditions

Average radon activity concentration derived from hourly measurements (Figure 3) was similar at both measurement locations, $15 \pm 11$ Bq m$^{-3}$ and $14 \pm 10$ Bq m$^{-3}$, measured in winter in Ljubljana and Vipava valley, respectively. Slightly lower average

$C_{Rn}$ was characteristic for spring: $13 \pm 9$ Bq m$^{-3}$ and $12 \pm 8$ Bq m$^{-3}$, respectively for both locations (Table 4). These values are above annual average outdoor radon concentration of 10 Bq m$^{-3}$ reported by UNSCEAR (2000) for the continental areas, which complies with the seasonal variation, which usually results in higher winter concentrations due to limited atmospheric mixing.

Apart from the changes in radon exhalation rate from the ground, time evolution of $C_{Rn}$ is mainly affected by atmospheric

dispersion characteristics. Periods of mechanically driven mixing within the PBL, with stronger wind speeds, and periods of prevailing thermally driven mixing can be distinguished especially during winter months, which results in irregular diurnal





time evolution. Typical diurnal variation is more pronounced in spring, when thermally driven atmospheric mixing prevails. Due to the limitations of the box model, this study is limited to the cases with thermally driven convective boundary layer. With this regard only days with average daily wind speeds below the 80$^{th}$ percentile of all values were considered and addressed further on as "normal" wind conditions. Local wind conditions are further presented in the Supplement (Section 3, Figure S4

and Figure S5). Diurnal variation of $C_{Rn}$ in thermally driven convective mixing, presented in Figure 4, reflects daily evolution of the PBL with the lowest $C_{Rn}$ in the middle of the day, when PBL is fully mixed. The lowest values of $C_{Rn}$ are on average around 5 Bq m$^{-3}$. $C_{Rn}$ starts to increase with the afternoon transition to the stable boundary layer and reaches highest values in the early morning. The amplitude of diurnal variation is controlled by PBL stability and duration of SNBL, resulting in the highest morning peak values in winter months, when NSBL regime lasts longer.

### 3.2  Black carbon concentration, diurnal and seasonal cycle

Clear seasonality of BC concentrations was observed at both urban background sites. Higher concentrations were measured in the colder season (Figure 5), resulting from weaker dispersion characteristics within the more stable PBL, as well as from stronger biomass burning sources. Despite significantly higher population density in LJ, BC concentrations are only about

25 % higher in LJ than in AJ.

The average BC concentration in winter was $4.5 \pm 5.7$ and $3.4 \pm 4.2$ µg m$^{-3}$ for LJ and AJ, respectively (Table 4). However, a significant micrometeorological difference between both locations has to be considered. Vipava valley is characterised by two extremes in atmospheric stability: very stable atmospheric conditions with strong pollution events can shift within few hours to the strong bora wind conditions, which disperse all atmospheric pollutants to the nearly regional background levels. In fact,

during stable PBL conditions in winter, BC concentration in AJ can easily exceed concentrations in LJ, reaching average daily BC concentration of $10 - 15$ µg m$^{-3}$. Based on the source apportionment model described in Section 2.2, there was a significantly higher contribution of biomass burning observed during winter in AJ (62 %) than in LJ (31 %), corresponding to the rural characteristics of Vipava valley area. Significantly lower BC concentrations were measured in spring, $1.5 \pm 1.6$ µg m$^{-3}$ and $1.1 \pm 1.2$ µg m$^{-3}$ in LJ and AJ, respectively.

BC concentrations at both hill sites were expectedly lower than that at the urban background sites. Golovec site (GOL), which is located 100 m above the city of Ljubljana is nevertheless more affected by urban emissions than Otlica site (OT), which lies about 830 m above the valley floor. BC concentrations measured at GOL were $2.2 \pm 2.0$, $1.1 \pm 1.0$ and $0.8 \pm 0.5$ µg m$^{-3}$ in winter, spring and summer, respectively, which is 51%, 42% and 38% lower than in the city. This indicates more intensive vertical dispersion of air pollutants (including BC) towards warmer season. Nevertheless, since vertical difference between

ARSO and GOL site is only 100 m, the GOL site remains above the ML only during very stable PBL conditions.

On the other hand, the vertical difference between the AJ and OT sites in Vipava valley is much larger, resulting in significantly lower BC concentrations on the hill. BC concentrations measured at OT during autumn, winter and spring season were similar, $0.4 \pm 0.5$, $0.6 \pm 0.8$ and $0.4 \pm 0.4$ µg m$^{-3}$, respectively. Slightly higher winter concentrations can be assigned to small local





contribution from a few houses which are spread on the slope around the observatory and small natural grass fire nearby the observatory on December 18, 2016, which increased BC concentrations to around 30 µg m⁻³ for several hours (Figure 5). The OT site is located above PBL most of the time during winter and therefore represents regional background BC concentrations. Towards spring, when MLH frequently reaches the OT site in the afternoon, the site is affected by polluted air masses from

the valley (Figure 9c) and BC concentrations increase. The OT site can lie during the night and morning hours in either the residual layer - in the case when MLH reached the OT site in the previous afternoon; or in free atmosphere - in the case when MLH remained below OT altitude in the previous day.

BC diurnal variation presented in Figure 6 reflects different dynamics of sources and their relative contribution. In general, the main driver of BC concentrations at both sites is atmospheric stability, leading to dispersion of pollutants during the day, and

subsequently lower BC concentration in the middle of day, and thus lower exposure of the population, except in the case of stable PBL conditions. Besides that, two peaks are usually observed in traffic related BC concentration ($BC_{TR}$), which is a combined consequence of traffic density and PBL stability. Morning traffic related peak of $BC_{TR}$ concentration is usually stronger at both locations, since dispersion of BC in the morning hours is limited due to low MLH. Due to higher traffic density and consequently stronger BC sources in LJ, it usually takes more time for BC to decrease during the day than in AJ. The

afternoon peak, on the other hand, strongly depends on daylight hours (which in general drives the PBL evolution). In winter, BC concentrations start to increase already between 16:00 and 17:00, whereas in spring and summer, much smaller increase can be observed, which is constrained to evening hours. Biomass burning BC sources are mostly limited to the colder season, when higher concentrations are measured especially in the evenings and first part of night. In contrast to LJ, AJ is affected by increased $BC_{BB}$ concentration also during the morning hours. During the weekends, lower $BC_{TR}$ concentrations are observed

in LJ, whereas no significant difference can be seen in AJ, leading to the assumption that meteorology plays much stronger role in Vipava valley than it does in Ljubljana (where BC sources are stronger) or it could be assumed that highway along the valley represents continuous source of BC, regardless of the week day.

### 3.3 Effective mixing height derived from box model

Hourly resolved MLH values were calculated based on Eq. 9, 11 and 12 for the whole period, when $C_{Rn}$ measurements were available. Although MLH results represent an intermediate model outcome and are actually not required for emission rate modelling, the results are important for understanding diurnal characteristics of PBL evolution and extent of pollutant dispersion, which allow us to compare the two locations from the point of view of micro-meteorological characteristics. They also serve as a quality control parameter of the model. Derived MLH for both locations, calculated from the specifically

selected monthly values of radon exhalation rate (Section 2.5) are presented on Figure 7. SNBL height was in general between 100 and 200 m a.g.l. at both locations and was found to slightly increase from cold to warm season. However, the seasonal pattern of SNBL height is not as pronounced as the seasonal pattern for the thermally driven daytime MLH. In February PBL reached its highest altitude at around 15:00, with the median MLH value for LJ of 450 m. On the contrary, in June MLH





reached its highest extent of 1550 m (median). In conditions of extremely unstable boundary layer, the maximum observed MLH extended higher than 2500 m at both locations. The influence of MLH on BC concentration measured at the urban background site (ARSO) and on the hill (GOL) is presented on Figure 8 for selected days, when derived MLH was validated from vertical profiles of BC concentration measured by plane. The highest BC concentrations at ARSO and the highest

difference between ARSO and GOL is observed during periods when MLH extends below the altitude of the hill site, 100 m a.g.l. (Figure 8 a).

The strongest PBL stability (excluding periods of Bora wind) in Vipava valley was observed in December 2016 and February 2017, when no significant diurnal variation could be detected. During these two months median MLH values at 15:00 were 230 m and 270 m, respectively. The highest vertical extent of PBL was observed in April, when MLH reached 1340 m

(median) at 16:00. Results of MLH values also explain the measured BC concentration in AJ and OT (Section 3.2), where comparison of BC concentration reveals the time periods, when both sites are located within the same air volume (i.e. periods when MLH overreaches OT site at 830 m a.g.l.). Especially in April and May, MLH reaches OT site in the afternoon frequently (from noon to 16:00), leading to increase of BC concentrations at OT and decrease of BC concentrations at AJ site (Figure 9 b and c). Good correlation was observed between MLH (derived from the box model) and vertically resolved lidar backscatter

signal over Vipava valley in periods when conditions are met for the application of both approaches. Figure 9 a represents measurements from January 9, 2017.

Results show that PBL reaches its full depth in the early afternoon, between 15:00 and 17:00, depending on the extent of daylight hours. Strong thermally driven mixing starts to diminish about 2 hours before sunset, followed by rapid transition to the SNBL conditions (Figure 7).

### 3.4 Black carbon emission rates

BC emission rates were determined in the second phase by inversion of BC concentrations based on the results of derived MLH values using Eq. 13 and 14. Emission rates from traffic ($E_{TR}$) and biomass burning ($E_{BB}$) sources were determined separately in order to take into account spatial characteristics and different dynamics of sources. Average daily BC emission

rates in different seasons for both locations are presented in Table 5. As expected, higher BC emissions are characteristic for Ljubljana, where overall BC emission rates ranged from 280 to 300 µg m$^{-2}$h$^{-1}$ in spring and winter, respectively. On the other hand, 25% lower overall BC emissions were found in Vipava valley and were in the range from 210 to 240 µg m$^{-2}$h$^{-1}$ in spring and winter, respectively. Emissions from traffic prevails in Ljubljana city and account for 80 % in winter time. On the other hand, biomass burning in individual houses contribute more than half (58 %) of the emitted BC in Vipava valley during

the heating season. In spring, however, outdoor temperature increases faster in the Mediterranean climate of Vipava valley than in does in Ljubljana, which means that heating season end much sooner. Similar biomass burning emission rates are thus characteristic for spring in Vipava valley and Ljubljana. The fraction of source specific emission rates slightly differs from the contribution fraction of actually measured BC concentrations from both sources (Table 4) after mixing and dispersion within


the PBL. Due to difference in daily dynamics of emission rate from biomass burning and traffic, the fraction of BC concentration from biomass burning is slightly higher than the fraction of its emission rate. Traffic emissions are occurring mostly during the daytime and are dispersed in the PBL more effectively than biomass burning emissions, with the sources active also during the night hours, thus having stronger impact on the concentrations. Average determined BC emission rates

are about two orders of magnitude higher than Slovenian national BC emissions of 2.2 kilotons reported by the EMEP (The European Monitoring and Evaluation Programme) emission inventory for 2016, which corresponds to the average hourly emission rate of $12.4\,\mathrm{\mu g\,m^{-2}h^{-1}}$. Difference is reasonable if we account for spatially heterogeneous emissions and the small contribution of less populated areas like forests and mountains. BC emission rates calculated in our study are nevertheless lower than reported for larger cities, as Kathmandu, where $316\,\mathrm{\mu g\,m^{-2}h^{-1}}$ and $914\,\mathrm{\mu g\,m^{-2}h^{-1}}$ were reported for summer and

winter period, respectively, or Delhi ($608\,\mathrm{\mu g\,m^{-2}h^{-1}}$) and Mumbai ($2160\,\mathrm{\mu g\,m^{-2}h^{-1}}$) (Mues et al., 2017).

Daily average $E_{TR}$ remains constant through the year and in general does not depend on outdoor temperature (Figure 10). Slight increase of $E_{TR}$ towards warmer days is observed in Vipava valley, which is probably a combination of several sources of uncertainty rising at the first place from the BC source apportionment model and uncertainties in the selection of radon exhalation rate combined with its daily changes, which are not accounted for in the model. Even a small difference in the BC

source apportionment can lead to underestimation of $E_{TR}$ and simultaneous overestimation of $E_{BB}$ in winter. As expected, $E_{BB}$ is higher in colder days due to the stronger heating demand. Since wood is the most frequently used fuel for heating in individual houses in the Vipava valley, emission rate increases much faster with colder days than it does in Ljubljana, where parts of the city dominated by individual houses are connected to the district heating system powered by the local thermal power plant. At both locations, higher $E_{BB}$ is observed when average daily temperature drops below 15 °C.

### 3.4.1    Hourly resolved source specific BC emission rate

Typical diurnal profile of $E_{TR}$ averaged over the whole measurement period reflects the traffic dynamics in the city of Ljubljana and in much smaller town – Ajdovščina in the Vipava valley (Figure 11). Minimum emissions are observed during night hours, between midnight and 4:00 in LJ and between 23:00 and 4:00 in AJ. Traffic and consequently $E_{TR}$ start to increase in the

morning around 5:00 and peak during working days between 7:00 and 8:00 with $E_{TR}$ of $280\,\mathrm{\mu g\,m^{-2}h^{-1}}$ and $250\,\mathrm{\mu g\,m^{-2}h^{-1}}$ (median values) in LJ and AJ, respectively. Morning peak is not observed during Sundays, when $E_{TR}$ during this period reaches significantly lower values: $60\,\mathrm{\mu g\,m^{-2}h^{-1}}$ and $110\,\mathrm{\mu g\,m^{-2}h^{-1}}$ in LJ and AJ, respectively. The morning peak is followed by a slight decrease of $E_{TR}$ in LJ, whereas in AJ, $E_{TR}$ drops substantially. $E_{TR}$ starts to increase again in the late morning and accelerates in the early afternoon at around 13:00. During working days, maximum $E_{TR}$ was observed during the afternoon

traffic peak, which is between 15:00 and 16:00 (median $E_{TR}$ of $530\,\mathrm{\mu g\,m^{-2}h^{-1}}$) in LJ, and between 15:00 and 17:00 in AJ (median $E_{TR}$ of $410\,\mathrm{\mu g\,m^{-2}h^{-1}}$). These results are comparable to results published by Ježek et al. (2018) for traffic BC emissions in Maribor (the second largest Slovenian city), where $E_{TR}$ during the afternoon peak in the range of $300 - 1300\,\mathrm{\mu g\,m^{-2}h^{-1}}$ is reported for 500 m × 500 m grid cells. In the evening hours $E_{TR}$ decreases faster in AJ than in LJ.





Since traffic BC emissions continue also after PBL shifts to the NSBL conditions, which is especially true during winter in LJ, this leads to stronger evening peak of $BC_{TR}$ concentration in LJ as compared to AJ. Sunday $E_{TR}$ was found to be higher in the afternoon, from 15:00 to 21:00 in Ljubljana (up to 340 µg m$^{-2}$h$^{-1}$) and from 16:00 to 18:00 in Ajdovščina (up to 240 µg m$^{-2}$h$^{-1}$), but it was nevertheless lower than during the working days. The emission rates are correlated with the traffic

density in Ljubljana (Figure S8).

Biomass burning BC emission rates, on the other hand, show weaker diurnal dynamics than traffic BC emission rates (Figure 12). Although seasonal variation of $E_{BB}$ is more pronounced, diurnal pattern in LJ shows slightly higher emission rates in the afternoon and evening hours, with hourly median $E_{BB}$ increasing from 10 µg m$^{-2}$h$^{-1}$ at 4:00 to 100 µg m$^{-2}$h$^{-1}$ between 15:00 and 19:00. In AJ, an additional morning increase is present in winter between 7:00 and 8:00, with $E_{BB}$ of 150 µg m$^{-2}$h$^{-1}$

followed by stronger afternoon peak of 420 µg m$^{-2}$h$^{-1}$ at 16:00. Although there were only 8 suitable days for model application in autumn, results show a similar diurnal pattern as in winter months. In spring, a similar afternoon $E_{BB}$ was obtained in LJ (60 µg m$^{-2}$h$^{-1}$) and AJ (70 µg m$^{-2}$h$^{-1}$).

It has to be pointed out, however, that higher uncertainty is expected for emission rate values in the afternoon hours, as discussed in Section 2.6.1. Horizontal advection, which is accounted for by introducing spatial decay constants to the box

model, can lead to overestimation of the emission rate in the presence of stronger wind conditions. Another contribution to the uncertainty arises from Rn measurements, which especially in strongly unstable PBL, reach the instrumental lower detection limit in the early afternoon. This can lead to underestimation of MLH and consequently also underestimation of derived BC emission rate. The comparison of the $E_{TR}$ and the traffic density in LJ shows the presence of the higher model uncertainty in the afternoon (Figure S8).

Negative hourly $E_{BC}$ values result from BC distribution, from temporal and spatial point of view, which does not comply with the expected background evolution of BC concentration. Thus, the local BC concentration peak, measured at the urban background site, would result in a high calculated emission rate, followed by negative calculation of the emission rate. The frequency of negative values also agrees with the time period when sources are active. Higher noise is thus obtained during unstable atmospheric conditions in the presence of local BC concentration peaks. The traffic BC emission rate thus results in

more noisy results than biomass burning BC emissions. Higher noise is also observed for the Vipava valley emission rate calculation, induced by stronger wind and non-homogeneous distribution of biomass burning sources. Negative values were treated as valid model results, since averaging results in a more realistic estimation of the emission rate.

## 4.    Conclusions

We present a method for the determination of the source specific black carbon emission rates and apply it to measurements in

two different environments: an urban location in Ljubljana and a rural one in the Vipava valley (Slovenia, Europe), which differ also in their natural characteristics (geology, geomorphology, meteorology). The influence of atmospheric dynamics was quantified based on atmospheric Rn concentration and monthly resolved $E_{Rn}$, allowing for 1-hour time resolution MLH



determination for periods of thermally driven PBL evolution. Intensity of BC sources – BC emission rate – was determined by taking into account horizontal advection term, simplified by temporal and spatial exponential decay. Whereas the choice of temporal decay constant introduces only small offset in determined BC emission rates, the spatial decay constant was shown to influence the daily pattern of calculated BC emission rates significantly. Different spatial decay rate was introduced for

traffic and biomass burning emission sources depending on the area under consideration and spatial distribution of both sources. Therefore $\gamma_{BB}$ for the Ljubljana basin was set to $5 \times 10^{-5}$ m$^{-1}$ which corresponds to half-distance decay of approximately 14 km. On the other hand, a smaller contributing area was chosen for $BC_{\mathrm{TR}}$, with $\gamma_{TR}$ set to $7 \times 10^{-5}$ m$^{-1}$ (corresponding to 10 km half-distance decay). Distribution of sources within the Vipava valley indicates smaller contribution area, with $\gamma_{BB}$ and $\gamma_{TR}$ set to $10^{-4}$ m$^{-1}$ (7 km half-distance decay).

The rural characteristics of Vipava valley area reflect in significantly higher BC contribution from biomass burning during winter in AJ (62 %) in comparison to LJ (31 %). The average BC concentration in winter was $4.5 \pm 5.7$ and $3.4 \pm 4.2$ µg m$^{-3}$ for LJ and AJ, respectively. However, during stable PBL conditions in winter, BC concentration in AJ can easily exceed concentrations in LJ, reaching average daily BC concentration of $10 - 15$ µg m$^{-3}$. BC concentrations decrease in warmer months.

Results show the overall BC emission rates in Ljubljana in the range from 280 to 300 µg m$^{-2}$h$^{-1}$ in spring and winter, respectively. Only 25% lower overall BC emissions were found in Vipava valley and were in the range from 210 to 240 µg m$^{-2}$h$^{-1}$ in spring and winter, respectively. This shows that the emission rates are not necessarily related to the population density and sparsely populated areas do feature high black carbon emission rates. As expected, BC emissions from traffic prevails in Ljubljana city and account for 80 % in wintertime. On the other hand, biomass burning in individual houses

contribute more than half (58 %) of the emitted BC in Vipava valley during the heating season. Due to the difference in respective daily dynamics of emission rates from biomass burning and traffic, the fraction of BC concentration from biomass burning is slightly higher than the fraction of its emission rate. Traffic emissions are occurring mostly during the daytime and are dispersed in the PBL more effectively than biomass burning emissions, with the sources active also during the night hours, thus having a stronger impact on the concentrations. Although BC concentrations from both sources decrease towards warmer

months, traffic related emission rates remain constant year-round, whereas biomass burning emission rates strongly depend on the outside temperature, which drives the heating demand.

Different diurnal pattern of $E_{\mathrm{TR}}$ was revealed for both measurement locations, reflecting traffic dynamics characteristic for Ljubljana and Vipava valley. Besides a narrow peak in $E_{\mathrm{TR}}$ in the morning (LJ: 280 µg m$^{-2}$h$^{-1}$ and AJ: 250 µg m$^{-2}$h$^{-1}$) the highest $E_{\mathrm{TR}}$ was observed during the afternoon traffic peak, which is between 15:00 and 16:00 (median $E_{\mathrm{TR}}$ of 530 µg m$^{-2}$h$^{-1}$)

in LJ, and between 15:00 and 17:00 in AJ (median $E_{\mathrm{TR}}$ of 410 µg m$^{-2}$h$^{-1}$). Biomass burning BC emission rates, on the other hand, show weaker diurnal dynamics than traffic BC emission rates. In Ljubljana, $E_{\mathrm{BB}}$ slowly increases from the early morning (10 µg m$^{-2}$h$^{-1}$) to the afternoon (100 µg m$^{-2}$h$^{-1}$). More pronounced daily dynamics of $E_{\mathrm{BB}}$ was observed in Vipava valley in winter with $E_{\mathrm{BB}}$ of 150 µg m$^{-2}$h$^{-1}$ between 7:00 and 8:00, followed by stronger afternoon peak of 420 µg m$^{-2}$h$^{-1}$ at 16:00.



Coupling of highly time-resolved measurements of primary, inert air pollutant, such as BC, with atmospheric radon concentration measurements provides a useful tool for direct, high time resolution measurements of intensity of emission sources. This information is essential for short-term forecast of air pollution episodes, as well as for the evaluation of the efficiency of air quality abatement measures.





**Figures**

Figure 1: Map of Slovenia (a) with marked areas of measurement sites Ljubljana (LJ) and Vipava valley (VV). b) area of Ljubljana with urban background (ARSO) and hill (Golovec – GOL) measurement sites. c) Area of the Vipava valley with urban background (Ajdovščina – AJ) and hill (Otlica – OT) measurement sites (Source: Map data ©2018 GeoBasis-DE/BKG (©2009) Google)





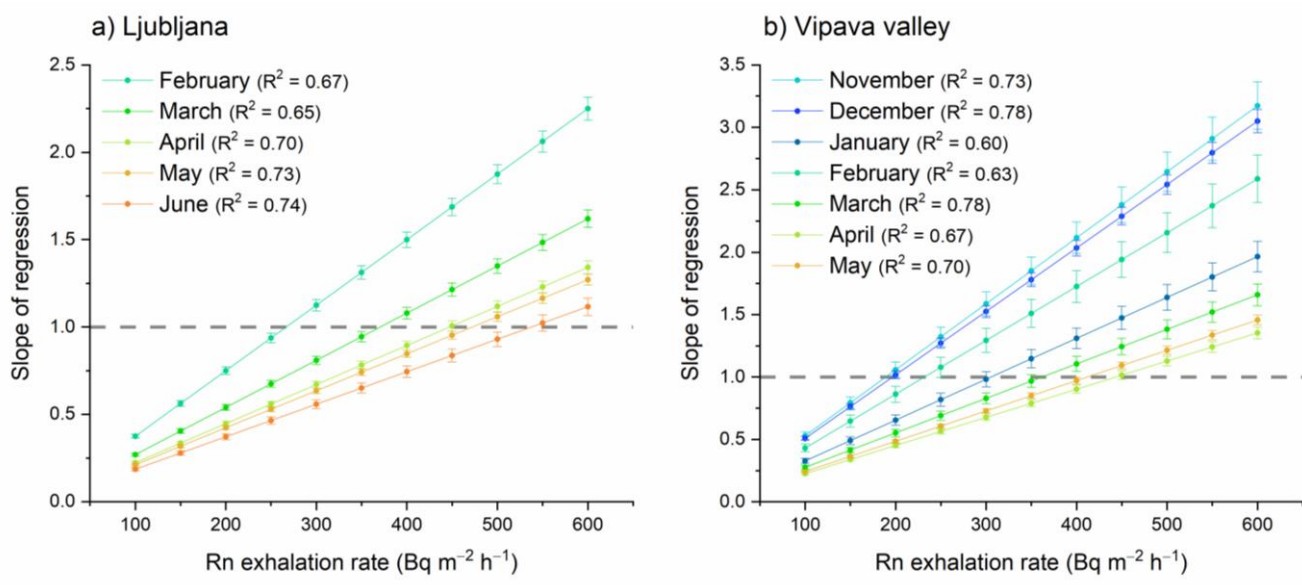

**Figure 2: Dependence of the slope of linear regression between calculated MLH from Rn measurements and GDAS data, on Rn exhalation rate for Ljubljana (a) and Vipava valley (b). This is used to determine the Rn exhalation rate at unity slope.**

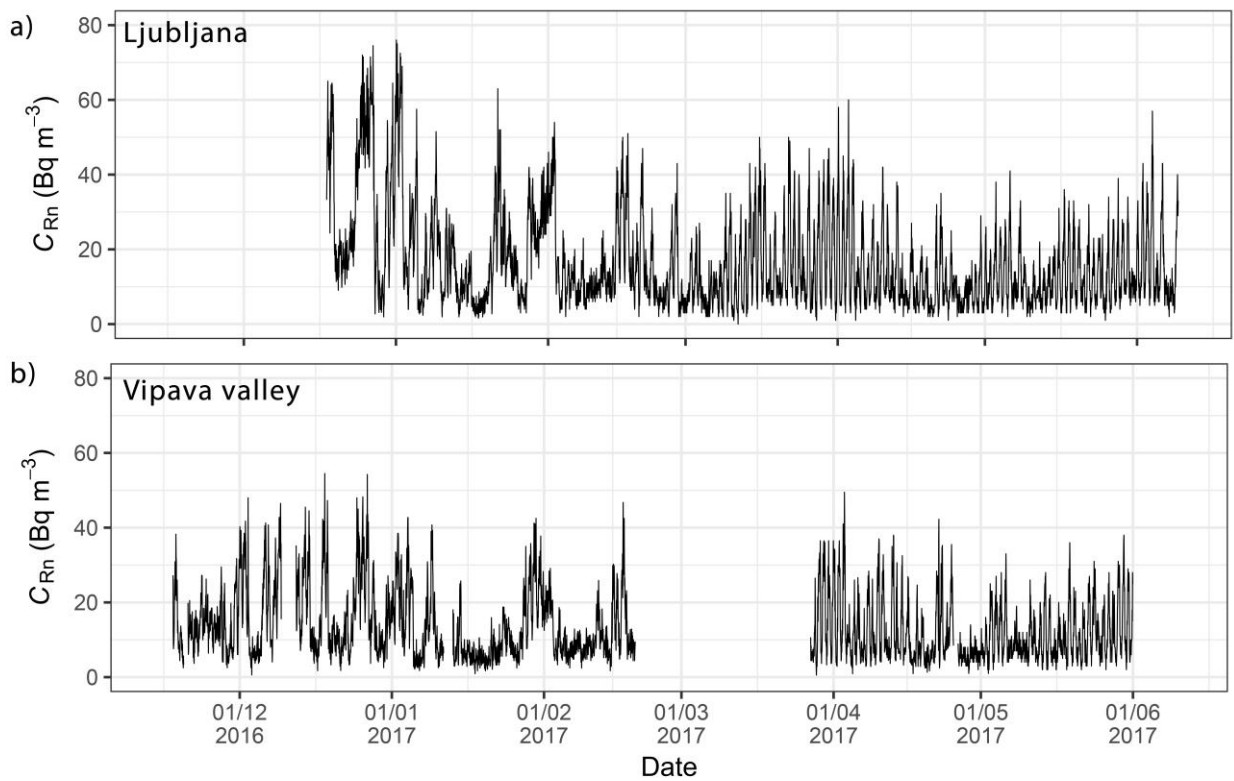

**Figure 3: Time series of radon activity concentration ($C_{Rn}$) measured in Ljubljana and (a) and in Ajdovščina (at the floor of Vipava valley).**



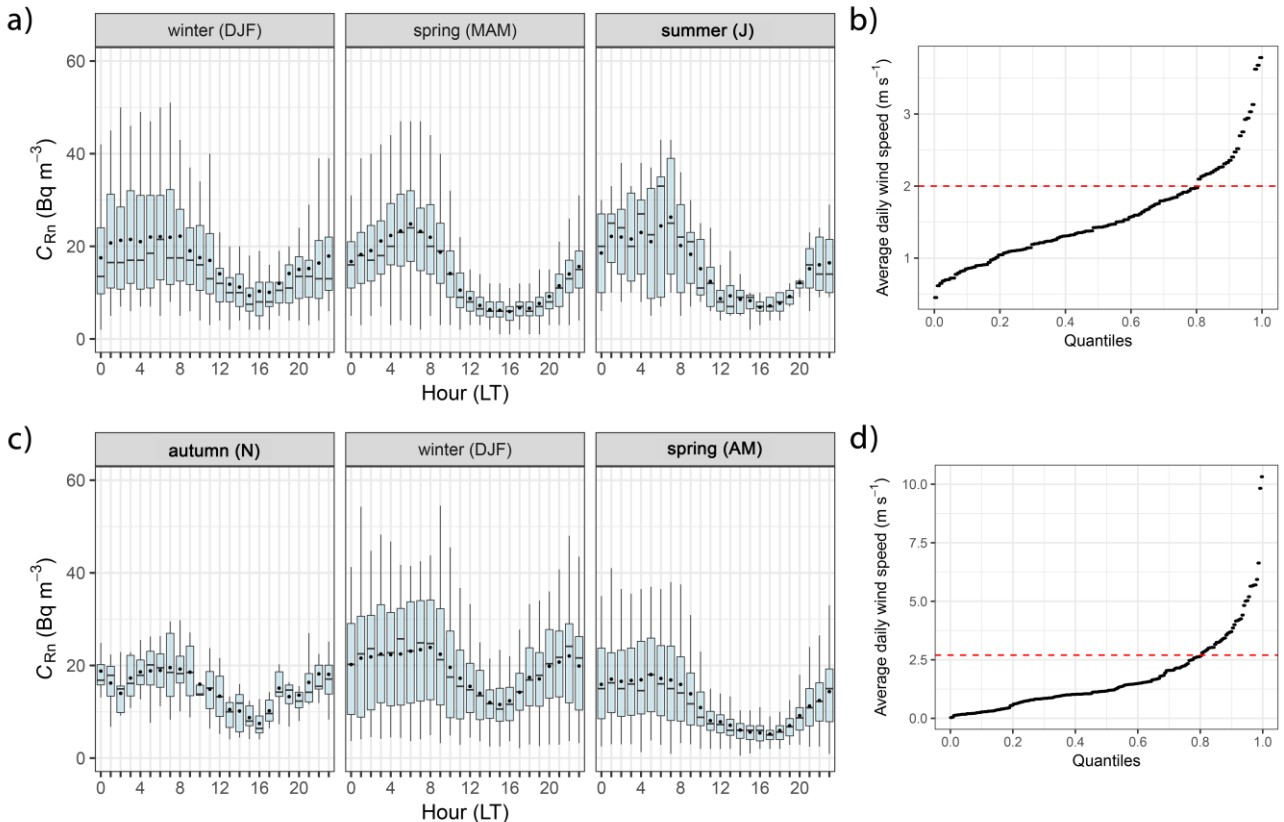

**Figure 4: Diurnal variation (local time: CET/CEST) of radon concentration ($C_{Rn}$) in Ljubljana (a) and Vipava valley (c), grouped by season for the whole period of Rn measurements. Statistics for every hour in a day are represented by a box plot derived from 1-hour data (point: mean, horizontal line: median, grey-coloured box: 25[th]-75[th] percentile, whiskers: 5[th]-95[th] percentile). Only days during which daily average wind speed is below the 80[th] percentile of all data (2 m s$^{-1}$ and 2.7 m s$^{-1}$ for Ljubljana and Vipava valley, respectively) are considered, as shown on plots b) and d).**



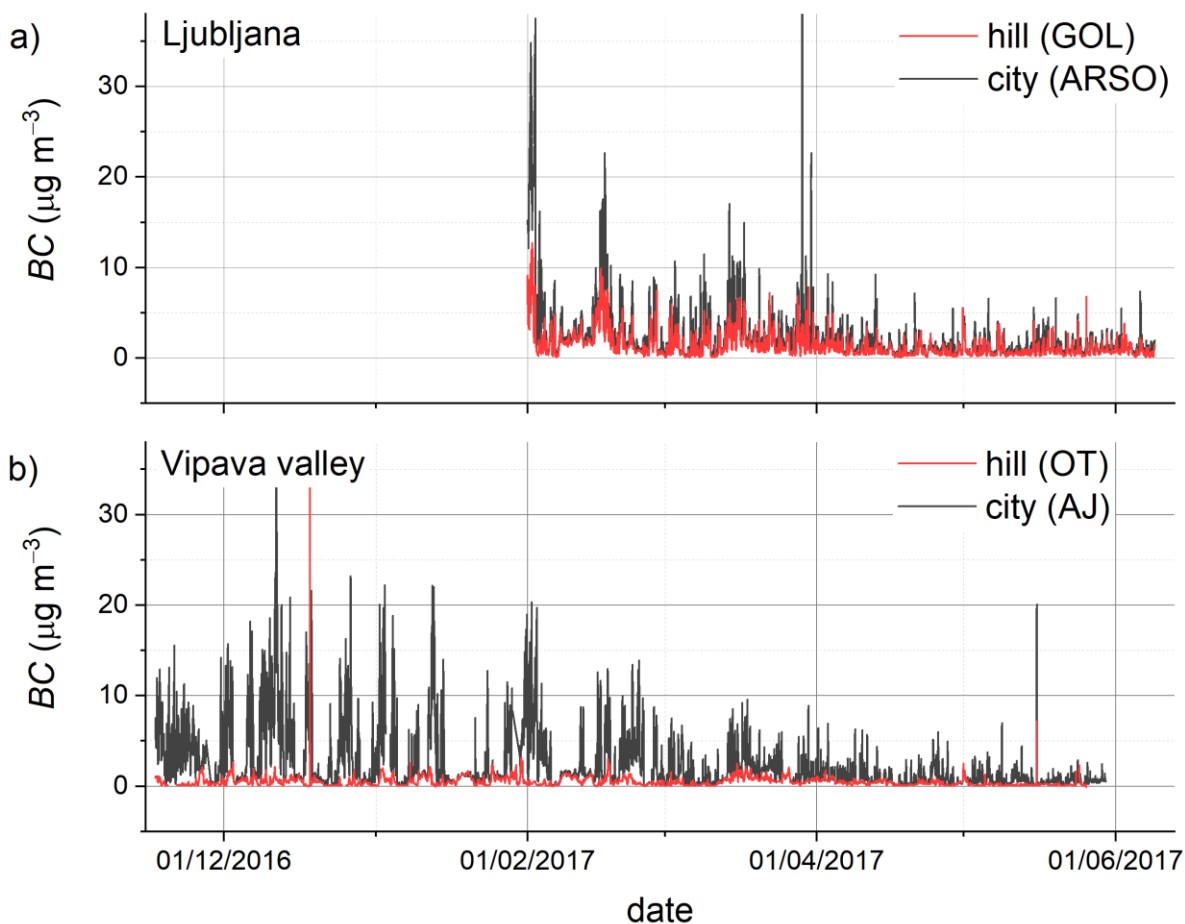

**Figure 5: Time series (1-hour running average is applied to 1-minute data) of black carbon concentration (BC) measured at two measurement sites (city - black, hill - red) in Ljubljana (a) and in Vipava valley (b).**

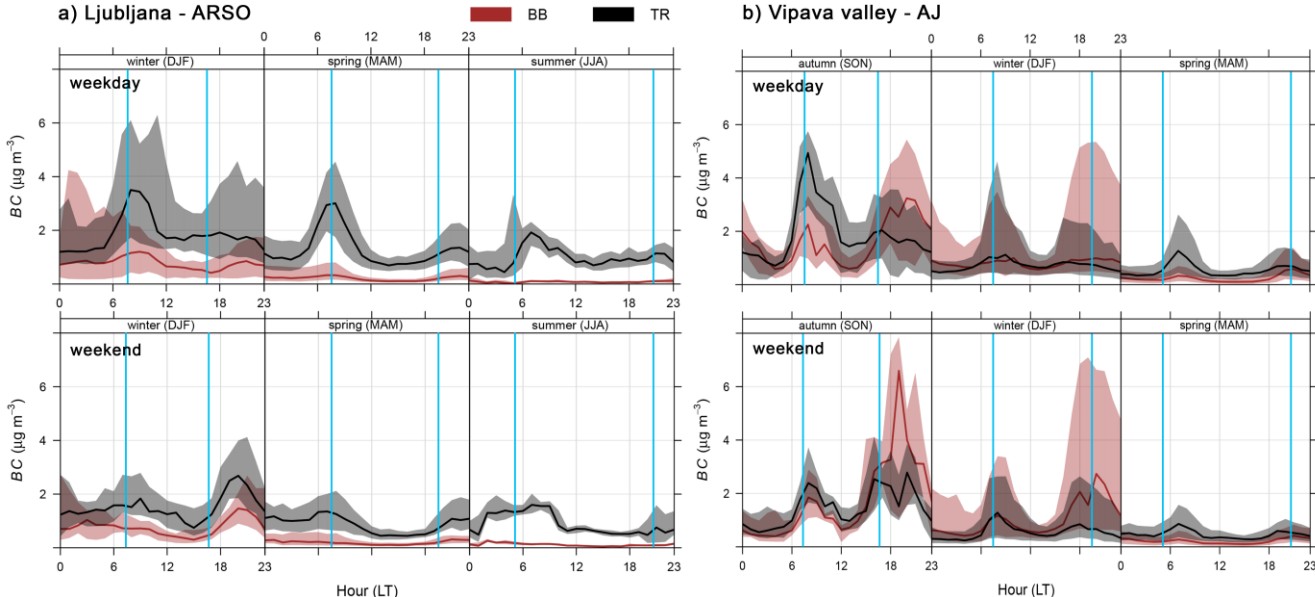

**Figure 6: Diurnal variation (local time: CET/CEST) of source specific black carbon concentration (traffic – TR and biomass burning – BB) at Ljubljana urban background - ARSO (a), and Vipava valley urban background - AJ (b), grouped by season and weekday/weekend. The statistics for every hour in a day are represented by median value (line) and 25th-75th percentile (shaded area) derived from 1-minute data. AAETR and AAEBB were set to 1 and 2, respectively. Blue vertical lines mark the sunrise and sunset time.**



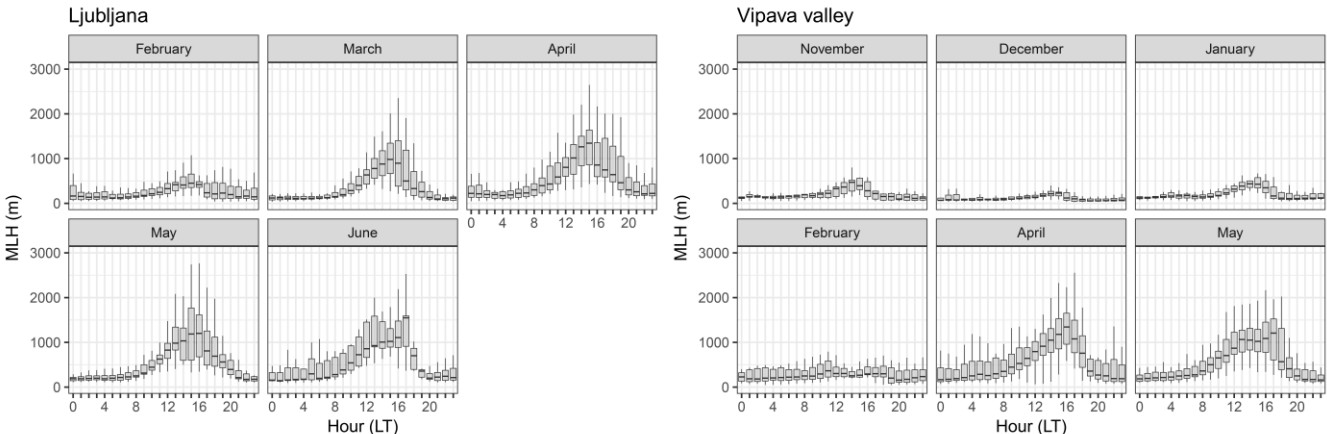

**Figure 7: Diurnal variation (local time: CET/CEST) of modelled mixing layer height (MLH) grouped by months for the periods of thermally driven PBL convection, for Ljubljana and Vipava valley. Hourly statistics are represented by box plots (horizontal line: median, grey-coloured box: 25th-75th percentile, whiskers: 5th-95th percentile).**



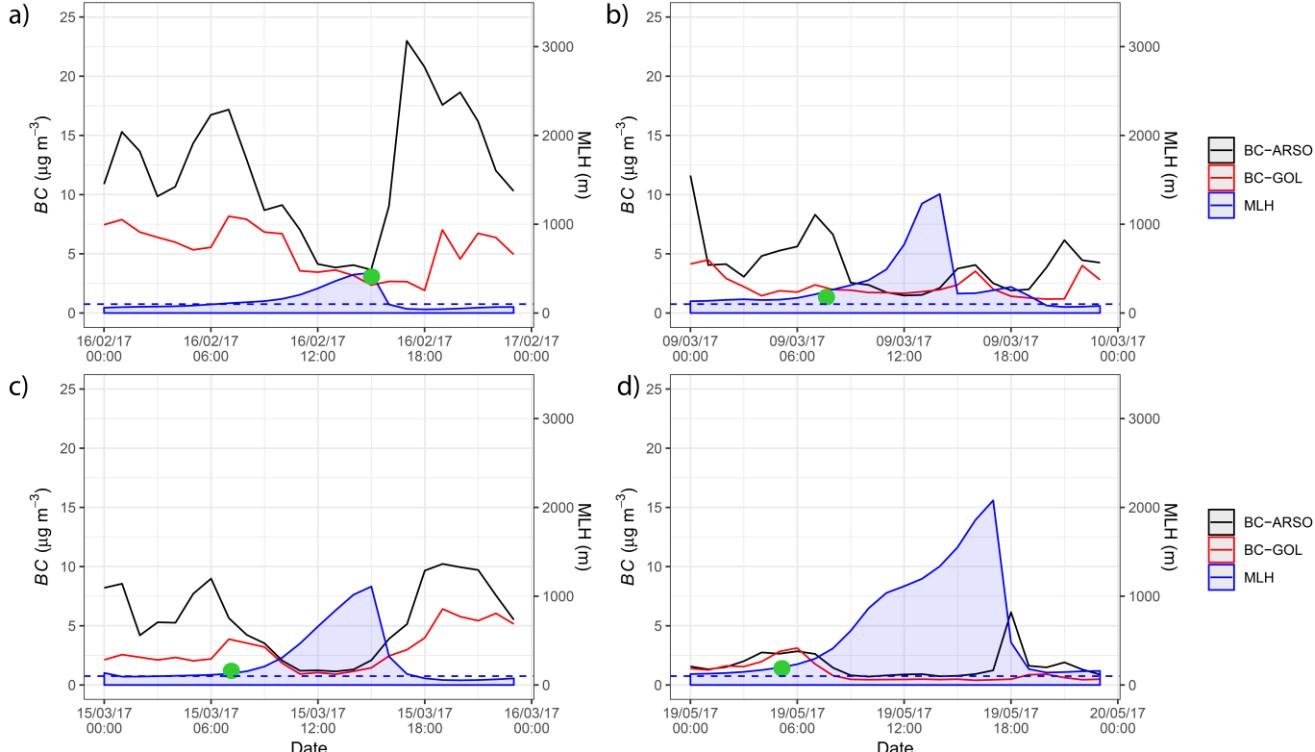

**Figure 8: Time series (UTC) of BC concentration measured in Ljubljana (ARSO) and on the hill (GOL) with modelled MLH and MLH determined by flight measurements (green point) on February 16, Feb 2017 (a), March 9,2017 (b), March 15, 2017 (c) and May 19, 2017 (d).**





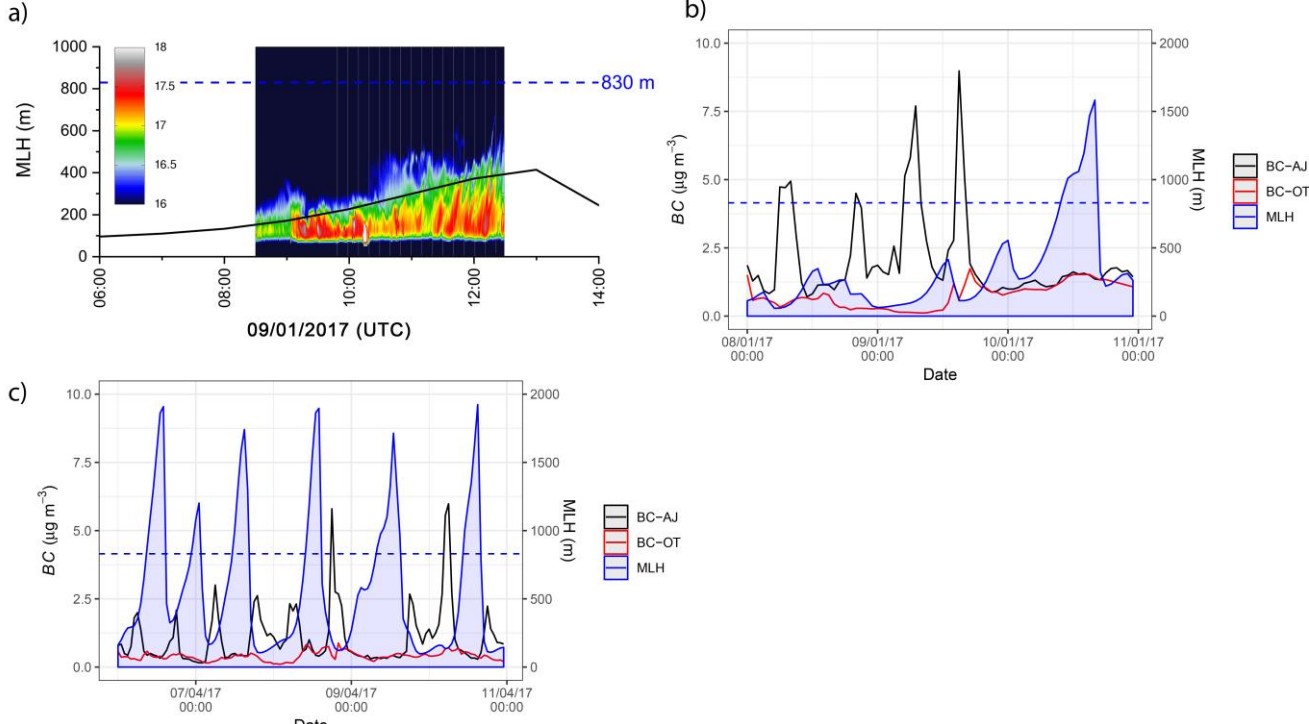

**Figure 9:** Comparison of modelled MLH (black line) over the Vipava valley with range-corrected lidar return signal on 09/01/2017 (a), time series (UTC) of BC concentration in the city (AJ) and on the hill (OT) with modelled MLH in the period January 8 – 10, 2017 (b) and April 6 – 10, 2017 (c). Dashed blue line represents the altitude of OT site, 830 m a.g.l. .





**Figure 10: Dependence of source specific emission rates $E_{TR}$ (traffic) and $E_{BB}$ (biomass burning) on the outdoor air temperature for Ljubljana (a, b) and Vipava valley (c, d). Daily average values are shown.**

## a) Ljubljana

## b) Vipava valley

**Figure 11: Diurnal variation (local time: CET/CEST) of emission rate of traffic related BC ($E_{TR}$) in Ljubljana (a) and Vipava valley (b), grouped by working days and Sundays. The statistics for every hour in a day are represented by a box plot (horizontal line: median, blue-coloured box: 25th-75th percentile, whiskers: 5th-95th percentile).**





**Figure 12: Diurnal variation (local time: CET/CEST) of emission rate of biomass burning related BC ($E_{BB}$) in Ljubljana (a) and Vipava valley (b), grouped by season (note the different scales). The statistics for every hour in a day are represented by a box plot (horizontal line: median, brown-coloured box: 25th-75th percentile, whiskers: 5th-95th percentile).**



**Tables:**

**Table 1: List of symbols and acronyms.**

| Acronym/Symbol | Definition | Units |
|---|---|---|
| $\lambda'_s$ | temporal decay constant | h$^{-1}$ |
| AAE | absorption Ångström exponent | |
| AAE$_{BB}$ | biomass burning related AAE | |
| AAE$_{TR}$ | traffic related AAE | |
| AJ | Ajdovščina | |
| ARSO | Slovenian Environmental Agency | |
| $b_{abs}$ | absorption coefficient | Mm$^{-1}$ |
| BC | black carbon concentration | ng m$^{-3}$ |
| BC$_{BB}$ | biomass burning related black carbon concentration | ng m$^{-3}$ |
| BC$_{TR}$ | traffic related black carbon concentration | ng m$^{-3}$ |
| C | multiple scattering parameter | |
| C$_{Rn}$ | radon activity concentration | Bq m$^{-3}$ |
| C$_s$ | species concentration | |
| E$_{BC}$ | black carbon emission rate | µg m$^{-2}$ h$^{-1}$ |
| E$_{BB}$ | black carbon emission rate from biomass burning sources | µg m$^{-2}$ h$^{-1}$ |
| E$_{TR}$ | black carbon emission rate from traffic sources | µg m$^{-2}$ h$^{-1}$ |
| EMEP | The European Monitoring and Evaluation Programme | |
| E$_{Rn}$ | radon exhalation rate | Bq m$^{-2}$ h$^{-1}$ |
| E$_s$ | species emission rate | |
| GDAS | Global Data Assimilation System | |
| GOL | Golovec Astronomical and Geophysical Observatory | |
| h | effective mixing layer height | m |
| LJ | Ljubljana | |
| ML | mixing layer | |
| MLH | mixing layer height | m |
| NOAA-ARL | NOAA Air Resources Laboratory | |
| OT | Otlica Meteorological observatory | |
| PBL | planetary boundary layer | |
| PM | particulate matter | |
| SNBL | stable nocturnal boundary layer | |
| T | air temperature | °C |
| wd | wind direction | |
| ws | wind speed | m s$^{-1}$ |
| $\gamma_s$ | spatial decay constant | m$^{-1}$ |
| σ$_{air}$ | mass absorption cross section | m$^2$ g$^{-1}$ |



**Table 2: Selected Rn exhalation rates ($E_{Rn}$) for each month for both measurement locations.**

| Month | $E_{Rn}$ (Bq m$^{-2}$ h$^{-1}$) | |
|---|---|---|
| | Ljubljana | Vipava valley |
| November 2016 | / | 200 |
| December 2016 | / | 200 |
| January 2017 | / | 300 |
| February 2017 | 250 | 250 |
| March 2017 | 350 | 350 |
| April 2017 | 450 | 450 |
| May 2017 | 450 | 400 |
| June 2017 | 550 | / |

5   **Table 3: Summary of temporal and spatial decay constants selected for Ljubljana and Vipava valley modelled area.**

| Measurement location | $\lambda$ (h$^{-1}$) | | $\gamma$ (m$^{-1}$) | |
|---|---|---|---|---|
| | TR | BB | TR | BB |
| Ljubljana | 0.006 | 0.006 | $7 \times 10^{-5}$ | $5 \times 10^{-5}$ |
| Vipava valley | 0.006 | 0.006 | $10^{-4}$ | $10^{-4}$ |

**Table 4: Summary statistics (mean ± standard deviation) of measured Rn concentration (Bq m$^{-3}$), BC (µg m$^{-3}$) concentration apportioned to traffic and biomass burning for urban background sites in Ljubljana (ARSO) and Vipava valley (AJ) and BC (µg m$^{-3}$)**
10   **concentration at the hill sites (GOL and OT).**

| season | Ljubljana | | | | | | Vipava valley | | | | | |
|---|---|---|---|---|---|---|---|---|---|---|---|---|
| | $C_{Rn}$ | $BC_{city}$ | $BC_{TR\text{-}city}$ | $BC_{BB\text{-}city}$ | BB % | $BC_{hill}$ | $C_{Rn}$ | $BC_{city}$ | $BC_{TR\text{-}city}$ | $BC_{BB\text{-}city}$ | BB % | $BC_{hill}$ |
| Autumn 2016 | / | / | / | / | / | / | 14 ± 7 | 3.2 ± 2.4 | 1.6 ± 1.4 | 1.6 ± 1.4 | 50 | 0.4 ± 0.5 |
| Winter 2016/17 | 15 ± 11 | 4.5 ± 5.7 | 3.1 ± 4.1 | 1.4 ± 1.8 | 31 | 2.2 ± 2.0 | 14 ± 10 | 3.4 ± 4.2 | 1.3 ± 1.6 | 2.1 ± 2.8 | 62 | 0.6 ± 0.8 |
| Spring 2017 | 13 ± 9 | 1.9 ± 1.9 | 1.5 ± 1.6 | 0.4 ± 0.5 | 21 | 1.1 ± 1.0 | 12 ± 8 | 1.1 ± 1.2 | 0.8 ± 0.9 | 0.4 ± 0.5 | 36 | 0.4 ± 0.4 |
| Summer 2017 | 16 ± 11 | 1.3 ± 0.9 | 1.2 ± 0.9 | 0.1 ± 0.1 | 8 | 0.8 ± 0.5 | / | / | / | / | / | / |





**Table 5: BC emission rates (mean ± standard deviation derived from daily mean values) for TR (traffic) and BB (biomass burning) emissions for each season and location, expressed in μg m$^{-2}$h$^{-1}$ and contribution of $E_{BB}$ to overall BC emissions.**

| season | Ljubljana | | | | Vipava valley | | | |
|---|---|---|---|---|---|---|---|---|
| | $E_{TR}$ | $E_{BB}$ | $E_{BB}/E$ (%) | # days | $E_{TR}$ | $E_{BB}$ | $E_{BB}/E$ (%) | # days |
| Autumn 2016 | / | / | / | / | 60 ± 80 | 50 ± 80 | 45 | 8 |
| Winter 2016/17 | 240 ± 170 | 60 ± 30 | 20 | 24 | 100 ± 80 | 140 ± 80 | 58 | 43 |
| Spring 2017 | 240 ± 110 | 40 ± 30 | 14 | 72 | 160 ± 70 | 50 ± 40 | 24 | 63 |
| Summer 2017 | 310 ± 130 | 20 ± 10 | 6 | 7 | / | / | / | / |





## Acknowledgement

This work was supported by TRL 6-9/4300-1/2016-60 grant from the Ministry of Economic Development and Technology of Republic of Slovenia and by research programmes I0-0033, P1-0385 and P1-0099 of the Slovenian Research Agency.

**Code/Data availability**

The data used in this publication is available upon request to the corresponding author (asta.gregoric@aerosol.eu).

**Author contribution**

AG, LD, IJ and GM designed the study, JV and AG performed and analysed radon measurements, ML, DG, LD and GM
performed and analysed measurements by ultralight aircraft, LW, MM and SS performed lidar measurements. Model development and paper preparation were performed by AG, LD and GM. All authors contributed to the scientific discussion.

**Conflicts of interest**

At the time of the research, A. Gregorič, G. Močnik and L. Drinovec were also employed by the manufacturer of the Aethalometer instruments, used to measure black carbon. Other authors declare no conflict of interest. The funding sponsors
had no role in the design of the study; in the collection, analyses, or interpretation of data; in the writing of the manuscript, and in the decision to publish the results.



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
