# Peer review of "The determination of highly time resolved and source separated black carbon emission rates using radon as a tracer of atmospheric dynamics"

_Atmospheric Chemistry and Physics, 2019_

## Referee Comment (RC1) · Scott Chambers (Referee) · 10 May 2020

11th May 2020

Dr. Barbara Ervens, Editor Atmospheric Chemistry and Physics

Dear Dr Ervens,

Thank you for the opportunity to review the discussion paper ACP-2019-911 "The determination of highly time resolved and source separated black carbon emission rates using radon as a tracer of atmospheric dynamics" by Gregorič et al., currently under consideration for publication in Atmospheric Chemistry and Physics. The paper is

dedicated to exploring and explaining differences in seasonal and diurnal source apportioned black carbon (BC) emissions between an urban and a semi-rural setting in Slovenia with strongly contrasting topographic settings. A particular focus of the paper involves the combined use of a box model (e.g. Williams et al. 2016; Salzano et al. 2016) and hourly Radon-222 observations to account for atmospheric dilution influences on BC concentrations, thereby enabling seasonal and diurnal estimates to be made of the separate emission fluxes of traffic-related and biomass burning-related BC. To my knowledge, inverting this kind of box model in order to obtain source apportioned BC emission rates has not previously been published.

Given the significance of BC in atmospheric particulate matter, both in terms of its potential climatic and health impacts, I believe that the study would be of interest to the readership of ACP. However, before I could recommend this manuscript for publication there are some fundamental issues pertaining to the analyses that would need to be addressed, potentially including a revision of the intended study scope. Together these changes would constitute major revision. I have noted my key concerns below.

Specific key concerns

1. Measurement heights and site characteristics: Much well-supported literature (including Karstens et al. 2015; ACP, 15, 12845-12865, 10.5194/acp-15-12845-2015), indicates that radon fluxes near Ajdovščina (AJ) are higher than near Ljubljana (LJ); by at least a factor of two. However, average radon (and its diurnal amplitude) reported in this study are higher at LJ than at AJ (Fig.4). This is likely attributable to: (i) a difference in radon sampling height (1m at LJ and 3m at AJ), (ii) the fact that the AJ radon observations were made on sloping ground, and (iii) the relative proximity of AJ to the coast ($\sim$20 km SW) and significant mountain peaks ($\sim$10 km N – NE). The greater measurement height at AJ would reduce observed radon concentrations cf. LJ (particularly at night), and the sloping terrain would contribute to frequent katabatic flow, which deepens the nocturnal boundary layer (further reducing concentrations), and reduces radon build up within the stable nocturnal boundary layer (SNBL) (since, the ultimate source

air for the katabatic flow is the lower troposphere, where radon concentrations can be very low). In support of this hypothesis, compare the spring diurnal cycles between sites (Fig. 4); AJ observations do not exhibit the distinctive pre-dawn radon peak seen at LJ characteristic of undisturbed accumulation in the SNBL.

Two assumptions of the box model employed in this study are: (i) a well-mixed SNBL, and (ii) a uniform radon source function within the region that could influence the model. In reality, concentration profiles within the SNBL exhibit strong gradients. Consequently, making a direct comparative analysis between sites where concentrations are recorded at different heights from the surface (without correcting for this) could introduce significant biases. Furthermore, while the radon source function near LJ would likely be uniform on spatial scales that influenced the model, this is not the case for AJ. The Adriatic coast lies ∼20km SW of AJ (beyond which the radon flux effectively drops to zero). Mountain peaks of >1000m lie ∼10 km N-NE of the site; at night under low to moderate wind speeds (as selected for this study), air would often be drawn from the lower troposphere, within which radon concentrations can also be very low.

In addition to the differences in radon sampling, BC observations at LJ and AJ were made at 4 and 20m a.g.l., respectively, at the primary sites. As mentioned above, at night under stable conditions, irrespective of potential differences in BC source strengths between the sites, or the flushing effect of katabatic flow at AJ, a significant gradient in BC concentrations would be expected in the SNBL between 4 and 20m agl. Even if both sites were on level ground, it would be necessary to estimate and correct for the separate sampling height differences between the sites before attempting a direct comparative analysis (at least at times when the ABL wasn't well mixed). The advective losses of radon in the SNBL at AJ are a separate complicating factor, and may change with wind direction. While this study excludes the highest 20% of wind speeds, all others are treated equally. In the related study of Williams et al. (2016), the atmospheric class typing approach employed selected several groups of mixing conditions each containing relatively consistent/similar wind speed and direction (which

reduced the uncertainty of the spatial decay constant estimates).

2. Unsuitable radon flux estimates: As noted by the authors (e.g. P5 L9; P8 L13-14), successful application of this box-model technique, and subsequent accuracy of the BC emission estimates, is contingent upon reliable knowledge of the radon flux at each site and its seasonal variability. On page 12 the authors outline the approach used to estimate the seasonality of radon fluxes at LJ and AJ. Contrary to existing literature, derived radon fluxes were found to be higher at LJ than at AJ. Furthermore, the reported seasonal variability of radon flux at LJ was from 70 - 150 mBq/m2/s, compared with existing literature estimates of 15 - 25 mBq/m2/s, yet the quoted uncertainty of the adopted flux estimation technique was $\pm$15 mBq/m2/s. Clearly, the derived radon fluxes are not appropriate for use in this study, and I would urge the authors to further investigate the cause of this discrepancy in flux estimates.

Radon fluxes were estimated by regressing mixing depths from the box model (using a range of assumed fluxes), against mixing depths from the NOAA-ARL GDAS database. More information about the data selection criteria for these regressions is warranted here (including an example regression plot). Even if only using fair-weather data it would not be appropriate to make these regressions using values across the whole diurnal cycle since (i) the radon / box-model mixing height estimates are most poorly defined for the 3-5 hours in the mid-afternoon when the GDAS data is most representative of "reality", and (ii) nocturnal mixing depths in the GDAS database are worst at night under stable conditions, when the radon / box-model method works best (in fact, the nocturnal GDAS data has a minimum reported value of 250m a.g.l. for nocturnal mixing under stable conditions; which is around a factor of 2 higher than corresponding nocturnal mixing depths predicted by the radon / box-model method). With this in mind, perhaps the mixing depth transition periods (e.g. between 7am and noon) would be best to use (if the resolution of the GDAS record was adequate)?

3. Afternoon box-model mixing depths: When the authors report whole (24-h) diurnal cycles of effective mixing depths based on the radon / box-model approach, further

discussion regarding the uncertainty of the mid-afternoon values is warranted. In my opinion, neither of the cited papers (Allegrini et al. 1994 or Vecchi et al. 2018) provide robust evidence for the efficacy of this mixing depth calculation approach under convective afternoon conditions. As noted by Williams et al. (2016), several hours after the onset of morning convection a number of the necessary assumptions for the box model approach are no longer valid, until convective mixing begins to decay again in the late afternoon. Typically, for 3-5 hours in the mid-afternoon hourly $\Delta Rn$ values that form the denominator of equations 9, 11 & 12 approach zero (absolute radon concentrations at this time were also often near the instrument's detection limit). In the mid-afternoon of convective days it is not clear that mixing-related influences on $\Delta Rn$ dominate over advective influences, and depending on the meteorological conditions of the prior several days, radon concentrations in the lower troposphere (that can be entrained to the ABL once the residual layer has been eroded) can vary by 2 orders of magnitude. Applying a low-pass filter to the radon record (with a $4 - 12$ hour cut-off; as done in this study and Vecchi et al. 2018) may improve the stability of the box model, but the actual ABL mixing characteristics at this time on a day to day basis are not correctly represented (since the variability being removed by the filtering process is a mixture of instrumental noise and several competing real physical influences). The largest BC ETR fluxes (with the largest uncertainties), are reported at these times (e.g. 2-4pm) for both workdays and Sundays – despite peak Sunday traffic not occurring at this time. Caution should be used when interpreting values at these times as they could bias daily averages.

The authors have sought to evaluate the fidelity of the radon / box-model's mixing depth estimates in two ways: firstly, using lidar observations (Fig. 9a), where results are very encouraging (for the chosen example) – although the comparison period ends around noon (near the time that the problematic afternoon period referred to above begins); and secondly, with vertical BC profiles recovered by drone. However, the chosen method to retrieve mixing depth estimates from the drone profiles appears to give inconsistent results. A visual inspection of Fig. S3 (a) indicates a well-mixed layer that terminates somewhere between $250 - 300$m agl, yet the profile analysis method

returns a value of 412m agl. A visual inspection of Fig. S3b suggests an inversion height roughly 250m agl., whereas the chosen analysis method returns an estimate of 181 m agl. Furthermore, the reported uncertainty for these profile-derived mixing depths is $\pm$1-3m, which is clearly unrealistic. If other parameters were retrieved from the drone (e.g. temperature, humidity or wind speed), these might help to improve the accuracy of the estimates.

4. Scope of investigation: Given the measurement complexities at the AJ site, and frequent failure of measurement conditions to satisfy necessary assumptions for application of the box model, if a more accurate estimate of the local radon flux can be made the authors might consider restricting the scope of their analysis of source apportioned BC emission rates to the Ljubljana region? There would still be sufficient interest and novelty in the results of such a study to warrant publication.

As an example of the influences of spatial heterogeneity of the radon flux near AJ, consider the wind speed threshold of $\sim$2.6 m/s set in this study to retain data for analysis. At 2.6 m/s, air masses arriving at the site in the afternoon from the southwest (Adriatic coast) may have radon concentrations of 0.5 – 1.0 Bq/m3 even for relatively shallow daytime mixing depths ($\sim$500m agl). On the other hand, air masses arriving during the afternoon from almost any other direction under comparable atmospheric conditions typically have radon concentrations of 5 – 10 Bq/m3. This change alone (unrelated to the ABL mixing depth) is around half the magnitude of the reported amplitude of the radon diurnal cycle.

---

## Referee Comment (RC2) · Anonymous Referee #2 · 28 Jun 2020

The manuscript by Gregorić et al. is an attempt to estimate Black Carbon emission rates on high temporal resolution over two contrasting environments. The manuscript presents a unique method of using Radon as a tracer to determine the mixing layer heights. The application of mixing layer height information and Spatio-temporal decay of Black Carbon concentration was further used to calculate the BC emission rates. The authors also compared their results with the BC emission estimates over some other regions of the world. The manuscript provides a valuable substitute for bottom-up approaches of estimating BC emission rates which are having high uncertainties in their activity, emission factors, and technology divisions. Although new, and limited over a few locations only, the pioneer method used in the manuscript could be useful

for other regions also. I recommend to accept the manuscript after resolving a few issues which are as follows:

1. The introduction section is unusually long with full of irrelevant information. I would suggest the authors modify the introduction section and re-write. Instead of describing the general impacts and roles of BC in the atmosphere (which are widely available in the literature), focus your description on the existing emission estimates, their problems, and the need for the methods which have been described in the manuscript. Two pages would be more than sufficient for the introduction.

2. Section 2.1 is also full of unnecessary information. Page 6, line 13-31: please reduce the content. There is no need to describe the population, growth, and implementation of various plans by the municipality. Please merge the 'measurement locations' and 'geological setting' together.

3. Page 8, line 3-30, already available in the literature, not needed specifically. Just cite the literature and remove the theoretical information.

4. Page 9, line 12: Please add the full form of FFT in the list of abbreviations in Table 1.

5. Section 2.4: It is recommended to provide a scatterplot of modelled-MLH with GDAS also in supplementary file.

6. Page 14, line 23: Authors mention that the Average Radon activity concentration was similar at both measurement locations, i.e., 15±11 Bqm-3 and 14±10 Bqm-3. At the same time, the authors also mentioned that it was slightly lower in the spring 13±9 Bqm-3 and 12±8 Bqm-3. Consider the standard deviations in the data, I do not see any difference in the data. Authors should check whether these differences are statistically significant or not and add a line on it.

7. Page 15, line 14: Despite significantly higher. . . . ... . .. . ..25% higher in LJ than in AJ. What is meant by only 25% higher? And how population is a factor here? It looks

highly ambiguous statement. The authors should consider removing it.

8. Figure 1 should be modified significantly. The background map items are almost invisible.

---

## Author Comment (AC1) · 15 Sep 2020

**Answers to Referee 1**

We would like to thank dr. Chambers (Referee 1) for helpful comments and recommendations for improving the manuscript. The comments have helped making the manuscript much more transparent and easier to read. We have carefully examined the comments, modified the manuscript, and extended the modifications to answer all open questions. The uncertainties of the method and weaknesses of the manuscript were carefully investigated, and discussion will be provided point by point to answer the concerns raised by dr. Chambers. The main considerations are grouped in the following points:

1. **Site characteristics**

- *"The greater measurement height at AJ would reduce observed radon concentrations cf. LJ (particularly at night), and the sloping terrain would contribute to frequent katabatic flow, which deepens the nocturnal boundary layer (further reducing concentrations), and reduces radon build up within the stable nocturnal boundary layer (SNBL) (since, the ultimate source. air for the katabatic flow is the lower troposphere, where radon concentrations can be very low)."*
- *"Furthermore, while the radon source function near LJ would likely be uniform on spatial scales that influenced the model, this is not the case for AJ. The Adriatic coast lies ~20km SW of AJ (beyond which the radon flux effectively drops to zero). Mountain peaks of >1000m lie ~10 km N-NE of the site; at night under low to moderate wind speeds (as selected for this study), air would often be drawn from the lower troposphere, within which radon concentrations can also be very low."*

2. **Measurement heights**

- *"Much well-supported literature (including Karstens et al. 2015; ACP, 15, 12845-12865, 10.5194/acp-15-12845-2015), indicates that radon fluxes near Ajdovščina (AJ) are higher than near Ljubljana (LJ); by at least a factor of two. However, average radon (and its diurnal amplitude) reported in this study are higher at LJ than at AJ (Fig.4). This is likely attributable to: (i) a difference in radon sampling height (1m at LJ and 3m at AJ), (ii) the fact that the AJ radon observations were made on sloping ground, and (iii) the relative proximity of AJ to the coast (_20 km SW) and significant mountain peaks (_10 km N – NE)."*
- *"Two assumptions of the box model employed in this study are: (i) a well-mixed SNBL, and (ii) a uniform radon source function within the region that could influence the model. In reality, concentration profiles within the SNBL exhibit strong gradients. Consequently, making a direct comparative analysis between sites where concentrations are recorded at different heights from the surface (without correcting for this) could introduce significant biases."*
- *"In addition to the differences in radon sampling, BC observations at LJ and AJ were made at 4 and 20m a.g.l., respectively, at the primary sites. As mentioned above, at night under stable conditions, irrespective of potential differences in BC source strengths between the sites, or the flushing effect of katabatic flow at AJ, a significant gradient in BC concentrations would be expected in the SNBL between 4 and 20m agl. Even if both sites were on level ground, it would be necessary to estimate and correct for the separate sampling height differences between the sites before attempting a direct comparative analysis (at least at times when the ABL wasn't well mixed)."*

3. **Selection of the suitable radon exhalation rate for the investigated region**

- *"On page 12 the authors outline the approach used to estimate the seasonality of radon fluxes at LJ and AJ. Contrary to existing literature, derived radon fluxes were found to be higher at LJ than at AJ. Furthermore, the reported seasonal variability of radon flux at LJ was from 70 - 150 mBq/m2/s, compared with existing literature estimates of 15 - 25 mBq/m2/s, yet the quoted uncertainty of the adopted flux estimation technique was ~15 mBq/m2/s. Clearly, the derived radon fluxes are not appropriate for use in this study, and I would urge the authors to further investigate the cause of this discrepancy in flux estimates. Radon fluxes were estimated by regressing mixing depths from the box model (using a range of assumed fluxes), against mixing depths from the NOAA-ARL GDAS database. More information about the data selection criteria for these regressions is warranted here (including an example regression plot). Even if only using fair-weather data it would not be appropriate to make these regressions using values across the whole diurnal cycle since (i) the radon / box-model mixing height estimates are most poorly defined for the 3-5 hours in the mid-afternoon when the GDAS data is most*

*representative of "reality", and (ii) nocturnal mixing depths in the GDAS database are worst at night under stable conditions, when the radon / box-model method works best (in fact, the nocturnal GDAS data has a minimum reported value of 250m a.g.l. for nocturnal mixing under stable conditions; which is around a factor of 2 higher than corresponding nocturnal mixing depths predicted by the radon / box-model method). With this in mind, perhaps the mixing depth transition periods (e.g. between 7am and noon) would be best to use (if the resolution of the GDAS record was adequate)?"*

- *"The authors have sought to evaluate the fidelity of the radon / box-model's mixing depth estimates in two ways: firstly, using lidar observations (Fig. 9a), where results are very encouraging (for the chosen example) – although the comparison period ends around noon (near the time that the problematic afternoon period referred to above begins); and secondly, with vertical BC profiles recovered by drone. However, the chosen method to retrieve mixing depth estimates from the drone profiles appears to give inconsistent results. A visual inspection of Fig. S3 (a) indicates a well-mixed layer that terminates somewhere between 250 – 300m agl, yet the profile analysis method. returns a value of 412m agl. A visual inspection of Fig. S3b suggests an inversion height roughly 250m agl., whereas the chosen analysis method returns an estimate of 181 m agl. Furthermore, the reported uncertainty for these profile-derived mixing depths is ~1-3m, which is clearly unrealistic. If other parameters were retrieved from the drone (e.g. temperature, humidity or wind speed), these might help to improve the accuracy of the estimates."*

**4. Model uncertainties in the mid-afternoon**

- *"When the authors report whole (24-h) diurnal cycles of effective mixing depths based on the radon / box-model approach, further discussion regarding the uncertainty of the mid-afternoon values is warranted. In my opinion, neither of the cited papers (Allegrini et al. 1994 or Vecchi et al. 2018) provide robust evidence for the efficacy of this mixing depth calculation approach under convective afternoon conditions. As noted by Williams et al. (2016), several hours after the onset of morning convection a number of the necessary assumptions for the box model approach are no longer valid, until convective mixing begins to decay again in the late afternoon. Typically, for 3-5 hours in the mid-afternoon hourly ΔRn values that form the denominator of equations 9, 11 & 12 approach zero (absolute radon concentrations at this time were also often near the instrument's detection limit). In the mid-afternoon of convective days it is not clear that mixing-related influences on ΔRn dominate over advective influences, and depending on the meteorological conditions of the prior several days, radon concentrations in the lower troposphere (that can be entrained to the ABL once the residual layer has been eroded) can vary by 2 orders of magnitude. Applying a low-pass filter to the radon record (with a 4 – 12 hour cut-off; as done in this study and Vecchi et al. 2018) may improve the stability of the box model, but the actual ABL mixing characteristics at this time on a day to day basis are not correctly represented (since the variability being removed by the filtering process is a mixture of instrumental noise and several competing real physical influences). The largest BC ETR fluxes (with the largest uncertainties), are reported at these times (e.g. 2-4pm) for both workdays and Sundays – despite peak Sunday traffic not occurring at this time. Caution should be used when interpreting values at these times as they could bias daily averages.*

**5. Scope of the investigation**

- *"Given the measurement complexities at the AJ site, and frequent failure of measurement conditions to satisfy necessary assumptions for application of the box model, if a more accurate estimate of the local radon flux can be made the authors might consider restricting the scope of their analysis of source apportioned BC emission rates to the Ljubljana region? There would still be sufficient interest and novelty in the results of such a study to warrant publication."*

We are aware, that further discussion on the uncertainties of the model and subsequent influences on the results is needed to properly present the results of this study. Therefore, a completely new section about the uncertainties has been added to the manuscript.

**Author's response**

The points described above will be addressed one by one and proposed changes to the Manuscript and Supplement will be provided under each point.

**1. Site characteristics**

The areas under investigation differ from the point of view of geomorphology and geology, as well as population density and main economic characteristics, the later affecting the pollution source density and activity. This is also the main reason for the selection of these two regions. As pointed out by dr. Chambers, the natural characteristics of Vipava valley may limit the application of the proposed method to shorter time periods, when model assumptions sufficiently satisfy the natural conditions at the site.

In this regard we investigated in detail characteristics of the wind field in Vipava valley and its effect on Rn concentration, to restrict model results only to the periods which meet the criteria of uniform radon exhalation rate. As noted by dr. Chambers, katabatic flow from the mountain ridge and winds from the Adriatic Sea may reduce Rn concentration in the valley.

Fig. 1 shows the wind characteristics of the Vipava valley as measured at the university building. In our reexamination and reanalysis of the data, for the reason of representativeness, we have used the meteorological data from the location of the BC measurements. At this point we have to apologize for the wrong statement in the first version of the manuscript, since wind measurements from Slovenian Environment Agency network were included, instead of wind data from the location of BC measurements.

Pronounced strong winds from NE can be observed on Fig. 1b, which corresponds to bora wind events – downslope wind blowing from Trnovo plateau towards the valley. Due to extreme conditions, which do not allow a buildup of local pollutant emissions, these events were excluded from the analyses. In the periods of weak to mild wind (below 2 m/s, Fig. 1 c) the wind flow follows the geomorphology of the valley: WNW-ESE. Winds in the direction from the Adriatic Sea (SW) are very rear, but it is true, that in rear situations, when SW wind with speed higher than 1.5 m/s is present, Rn concentration in the afternoon decreases to $2 - 3$ Bq/m$^3$ compared to $\sim 5$ Bq/m$^3$ for airflow from other directions (Fig. 1d). The Rn diurnal profile is similar in other wind directions and it only shows slight dependence on wind direction when the speed exceeds 1.5 m/s. The most pronounced change in Rn concentration is present with wind direction from NE (when wind speed exceeds 1.5 m/s), e.g. from the mountain ridge. However, the radon concentration does not decrease in these situations, but is instead higher ($\sim$ 10 Bq/m$^3$) than from other directions, probably due to higher radon source over the carbonate rocks of the Trnovo plateau. Higher Rn concentration would in turn decrease calculated MLH, eventually resulting in an underestimation of black carbon emission rates. No significant dependence of radon concentration on wind direction can be seen for pre-dawn period with low wind speed conditions.

Based on these observations, we have decided to limit the analyses at AJ location to the cases with average daily wind speed lower than 2 m/s (instead of previous 2.7 m/s), which accounts for around 60% of the measured dataset. Although at the higher end of wind distribution, we can expect some influence on the model results, periods with optimal atmospheric conditions can in our opinion still provide important information about black carbon emission rates.

Fig. 4 in the manuscript and Fig. S4 and S5 of the Supplement will be corrected accordingly.

a)

b)

[Figure]

c)

d)

**Figure 1: Vipava valley location: a) Digital elevation model of the Vipava valley with marked measurement sites, b) monthly wind rose, c) monthly wind rose for a subset of wind speed below 2 m/s, d) polar annulus plot shows how Rn concentration (color code in Bq/m³) vary by wind direction and hour of day .**

**2. Measurement heights and influence on the estimated radon exhalation rate**

We thank dr. Chambers for expressing the concern regarding measurement height and its influence on measured radon concentration and bias introduced by direct comparison of both sites. We are aware that this difference was not clearly presented and commented on the manuscript. Therefore, further clarification is necessary.

As explained by dr. Chambers, radon concentration profiles in the stable nocturnal boundary layer (SNBL) exhibit strong gradients. As shown for example by Ochmann (2005), measurements become characteristic for larger area with increasing height from the ground. The daily amplitude of radon concentration will thus be higher when measurements are conducted closer to the ground, especially in the SNBL conditions, which also affects an hourly increase of radon concentration ($dC_{Rn}$) in SNBL conditions

If we consider the case of stable atmospheric layer with known $h_i$, radon concentration measurements at different height above ground result in a different radon exhalation rate ($E_{Rn}$) estimate (Eq. 9 in the manuscript).

$$h_i = \frac{E_{Rn}\Delta T_{Rn}}{dC_{Rn}}$$

This actually means, that measured or modelled values (from existing databases) of radon exhalation rate has to be evaluated/calibrated based on the radon measurement height. The radon exhalation rate estimated in this study actually represents an effective $E_{Rn}$, rather than real exhalation rate ($E_{Rn-0}$) for the area under consideration.

$$E_{Rn} = E_{Rn-0} \times s,$$

where $s$ represents a scaling factor depending on the measurement height. This explanation was already introduced by Griffiths et al. (2013) in the study where mixing layer height from a radon model was compared with lidar derived mixing layer height. The scaling factor and thus the effective $E_{Rn}$ will be higher when measurements are performed closer to the ground. In the mentioned study, the average scaling factor was 1.87 for measurements at 2 m a.g.l. Although a known issue, it seems to be neglected in later publications (e.g. Williams et al., 2016; Vecchi et al., 2018).

As explained in the section 2.5 (Page 12), different measures of the mixing layer height were used for the estimation of $E_{Rn}$. This paragraph was updated in order to clarify the difference between the effective and real $E_{Rn}$ and explain the calibration procedure and how the effective exhalation rate is used in our model.

**Changes in the manuscript:**

**Page 12, lines 11 - 29**: (text marked in blue was added/deleted).
* * *
The results of MLH values, determined by the box model, strongly depend on the correct estimation of radon exhalation rate. As discussed in previous sections, $E_{Rn}$ is affected by seasonal meteorological changes mostly by varying soil humidity and permeability. Since continuous monitoring of $E_{Rn}$ is usually not available, the box model has to be calibrated to any the available information of MLH. Due to vertical gradients in radon concentration, which are present especially during the SNBL conditions, the height of radon measurements above ground level can play an important role in its observed daily variation, potentially biasing the results of the modelled MLH, if the measurement height is not taken into account. As introduced by Griffiths et al. (2013), the actual radon exhalation rate ($E_{Rn-0}$) has to be calibrated based on the radon measurement height. Therefore, the radon exhalation rate estimated in this study represents an effective $E_{Rn}$, rather than actual $E_{Rn-0}$ for the area under consideration:

$$E_{Rn} = E_{Rn-0} \times s \tag{13}$$

where $s$ represents a scaling factor which depends on the measurement height. Lower measurement height results in larger $s$. We use this effective exhalation rate, representing a wider region, in our model.

The calibration of the radon box model in terms of selection of appropriate $E_{Rn}$ was performed by combining three different approaches.

1. Comparison of the radon derived MLH (for $E_{Rn}$ in the range from $50 - 400$ Bq m$^{-2}$h$^{-1}$) based on Eqs. 9, 11, 12, with modeled values of MLH, obtained from the Air Resources Laboratory (NOAA-ARL) Global Data Assimilation System (GDAS) database. The approach is explained in detail in the Supplement Section 5.1. 100 Bq m$^{-2}$h$^{-1}$ acceptable range was used.

2. MLH determined in the first step was compared to black carbon measurements at different elevations. When MLH exceeds the elevation of the higher BC measurement site (hill), BC concentration is expected to be similar at both measurement sites (city and hill), whereas in the period when MLH is below the hill measurement site, a strong gradient in BC concentration is observed.

3. In the third step, radon derived MLH for selected days was compared to the MLH determined from vertical profiles of BC measured with an aircraft over the Ljubljana basin (Supplement Section 2) or with lidar-derived MLH in the Vipava valley.

The results of the first approach were used as the first estimate of the $E_{Rn}$. In the case of high uncertainty (low number of data points, wide confidence interval level, unrealistic values of $E_{Rn}$ estimate), the second approach was used to confirm the previously obtained $E_{Rn}$ estimates or to obtain a suitable range of $E_{Rn}$. For the months, when the BC vertical profiles or lidar-derived MLH were available, the third approach was used to estimate the $E_{Rn}$.

Graphical presentation of above-mentioned approaches for each month is presented in the Supplement Section 5.2.

This method allowed us to obtain the average monthly exhalation rate (Table 2) using the data from the meteorological model (GDAS), even though the model spatial and time resolution is low. We interpret this effective exhalation rate to be representative of the investigated regions for the purpose of using it in our model.

**Changes in the Supplement**

The description of the approach used to compare radon derived MLH with modelled values of MLH, obtained from the NOAA database was added to the Supplement (Section 5).
Scatter plot of MLH daily averages with the slope of regression, as well as time series of BC concentration in the city and hill site and validation with lidar or aircraft measurements was added to the supplement (Fig. S9 and S10). The final estimate of $E_{Rn}$ will be presented in a table in the main manuscript.
* * *
**5. Estimation of the radon exhalation rate**

**5.1 Calibration of $E_{Rn}$ with the NOAA mixing layer height**

The following procedure was used to compare radon derived MLH and MLH from the NOAA database. This was the first step for the estimation of appropriate monthly resolved $E_{Rn}$.

1. A subset of the dataset with daily average wind speed below 2 m/s limited to the period without rain was used for comparison.
2. In order to avoid the period of day, when both models have the highest uncertainty, only 9 hours from 4:00 – 13:00 were considered. Due to 3 hour time resolution of NOAA database, 4 points per day were included in the analyses, namely at 4:00, 7:00, 10:00, 13:00 (CET). A daily average (and standard deviation) of both MLH estimates was used for further analyses (Fig. S7 a).
3. Deming regression (Cornbleet and Gochman, 1979) was applied to obtain the regression slope between MLH from radon data and NOAA MLH data ($Z_i$) (regression was forced through zero) (Fig. S7 b). Deming regression minimizes the sum of distances in both the x and y direction. Standard deviation in x and y were used with the confidence level of 95%. Consequently, days with more stable atmospheric conditions have higher influence on the slope of regression.
4. The $E_{Rn}$ which resulted in the slope of unity was considered as the most appropriate $E_{Rn}$ estimate. $E_{Rn}$ was rounded to 50 Bq m$^{-2}$h$^{-1}$ with 100 Bq m$^{-2}$h$^{-1}$ acceptable range.

[Figure]

**Fig. S7 Example of a subset of data for Ljubljana from 19 – 24 March, 2017. Data points for 4:00, 7:00, 10:00 and 13:00 (CET) (matching the 3 hour time resolution of NOAA dataset) were considered. MLH obtained from the radon model is presented by a point and range calculated for 100 Bq m$^2$ h$^{-1}$ range (200, 250 and 300 Bq m$^2$ h$^{-1}$ are used in the selected case). Data for March 20 are omitted from the comparison due to average daily wind speed exceeding the value of 2 m s$^{-1}$. b) Deming regression is fitted through daily averages (and standard deviation) of MLH. Red dotted line represents the lower and upper 95% confidence interval, dashed grey line represents 1:1 line.**

**5.2 Monthly estimates of radon exhalation rate**

Appropriate $E_{Rn}$ was determined based on combination of three different approaches, as explained in Section 2.5 of the main text. Selected cases are presented for each month for LJ and AJ location.

1.  The slope of unity between radon based MLH and NOAA MLH data obtained by the procedure explained in the section Supplement 5.1, represents the first best estimate of $E_{Rn}$ (Fig. S8). In the case of high uncertainty (low number of data points, wide confidence interval range, unrealistic values of $E_{Rn}$ estimate), the second approach was used to confirm the previously obtained $E_{Rn}$ estimates or to obtain a suitable range of $E_{Rn}$.
2.  In the second phase estimated $E_{Rn}$ was evaluated by comparison of black carbon concentration measured at different elevation (city – hill). Strong vertical gradient in concentration indicates period when MLH is below the upper measurement site (hill).
3.  Lastly, radon based MLH was compared to the MLH estimated from the vertical BC profile measured by light aircraft in LJ and with lidar measurement for AJ location. When the data were available, the third approach was used superior to the approach 1 and 2.

a)                                                            b)

[Figure]

**Fig S8: Dependence of the slope of the Deming regression between calculated MLH from Rn measurements and NOAA data, on Rn exhalation rate for Ljubljana (a) and Vipava valley (b). This is used to determine the Rn exhalation rate at unity slope in the first phase. Points are connected with line for visualization purposes only. Dotted lines represent the envelope of 95% confidence interval.**

**BC sampling height:**

As mentioned in the manuscript and noted by dr. Chambers, black carbon measurements were conducted at different heights. Unfortunately, there was a mistake in the original submission in the sampling height at AJ location. After repeated measurements, we confirmed, that the sampling height was 12 m above ground, rather than 20 m. Despite this, stratification of SNBL can play a role in measured black carbon concentration, especially when comparing traffic and biomass burning BC sources, since their emissions have different characteristics. $BC_{TR}$ is emitted from the sources at the ground surface, whereas biomass burning sources are usually higher, at the height of the chimneys. Due to dispersion and dilution processes, the concentration gradient flattens with time. This means that in the case of traffic related BC, with sources mostly active during the day, there would be enough time for the dispersion of $BC_{TR}$ before stratification of SNBL. However, the morning emissions before the break of SNBL could be slightly underestimated in AJ, where sampling was performed at 12 m above ground. On the other hand, biomass burning BC is emitted higher above ground. For Ljubljana basin it was shown that $BC_{BB}$ is homogeneously distributed within the city (Ogrin et al., 2016). At AJ location, with numerous biomass burning sources over smaller area, the concentration profile could be more significant, especially in stable atmospheric conditions, which could lead to increased $BC_{BB}$ measured on the level of emissions. This could lead to slight overestimation of $BC_{BB}$ emission rate in the afternoon – early evening, before the emission plume is dispersed. We thank dr. Chambers to pointing out this difference, which was not discussed in the submitted version of the manuscript. Discussion regarding the influence of BC measurement height is added to the section at the end of the manuscript: 5.3 Uncertainty estimation.

**3. Selection of the suitable radon exhalation rate for the investigated region**

Dr. Chambers pointed out the unsuitable radon exhalation rate estimates used for both investigated locations. In this regard, we would like to point out two things:

1. Larger effective $E_{Rn}$ estimated for LJ than for AJ locations is partially explained by the difference in measurement heights, which affect the $E_{Rn}$ values (scaling factor mentioned above). The scaling factor reported by Griffiths et al. (2013) was 1.87 for 2 m a.g.l.

2. Modelled values of radon exhalation rate on the European scale, as published in Karstens et al. (2015) and Lopez-Coto (2013) are lacking the spatial resolution to be used as a reliable radon source term in the investigated area. Due to complex geology, the resolution may be too low to distinguish between local features of radon fluxes, which mainly follow the geological units (see the study of radon emanation of Slovenian soils by Kardos et al. (2015)). Flysch sedimentary rocks and carbonate rocks, which are completely different from the point of view of radon exhalation, are not clearly distinguished on the European scale, since flysch rocks follow the course of the valley, whereas in NE-SW direction, the lithology varies with spatial resolution below 3 km.
   As reported by Karstens et al. (2015), the radon exhalation varies in the range from 70 – 140 Bq m$^{-2}$h$^{-1}$ and 160 – 180 Bq m$^{-2}$h$^{-1}$ for AJ in winter and summer, respectively. The range of $E_{Rn}$ in LJ area is 40 – 90 Bq m$^{-2}$h$^{-1}$ and 90 – 130 Bq m$^{-2}$h$^{-1}$ for winter and summer, respectively. Although the values reported for the region of Vipava valley are larger than for the central part of Slovenia, the flysch and carbonate bedrock is not reported separately. Therefore, caution is necessary when using modelled values of $E_{Rn}$ on the European level to interpret local characteristics of investigated area and for the case of complex geological settings encountered in the Vipava valley, this approach would not be appropriate.

It still seems that after calibrating the radon model, the estimated radon exhalation rate (with scaling factor included) is too high for both sites. If a an approximate scaling factor of 2 would be considered for LJ location, the real $E_{Rn-0}$ would be in the range from 125 – 275 Bq/m$^2$h, which is approx. by a factor of 2 higher than the expected range of $E_{Rn-0}$ for Ljubljana area. For AJ location, with lower scaling factor – approx. 1.5 (due to measurements performed at 4 m height), $E_{Rn-0}$ would be in the 130 – 270 Bq/m$^2$h range. Therefore, we investigated in detail all the variables, which may lead to these results. By applying different approximations to the radon box model, the estimated radon exhalation rate was lowered to more realistic values. Details of the box model and changes applied are presented in the following section.

**4. Model assumptions and uncertainties in the mid-afternoon**

After investigating the reason for high radon exhalation rate estimates, the following observation was found: as explained in the manuscript (page 12, lines 2-10) and pointed out by dr. Chambers, due to unstable model results with high uncertainty of the effective MLH in the mid-afternoon, the Rn concentration in the residual layer was set to zero (***Approach A***) in the submitted version of the manuscript. With this estimation, the effective MLH

followed flatter curve during the morning transition from SNBL to daytime boundary layer (Fig. 2). When compared with MLH obtained from other measurements (NOAA, BC concentration profile, lidar), the best estimate of $E_{Rn}$ was higher than the expected range.

We have reconsidered the approximation of Rn concentration in the residual layer in order to obtain more realistic daily evolution of MLH. In the published studies of radon box model approach (e.g.Williams et al., 2016; Vecchi et al., 2018), Rn is considered to be stable in the boundary layer (**Approach B**), affected only by the rate of its radioactive decay. The calculated MLH, however, results in strong overestimation of the mid-afternoon MLH, when PBL is fully developed.

Since Rn concentration in the residual layer actually decreases with height (Griffiths et al., 2011), a linear decrease of Rn concentration in the residual layer was finally used in the model (**Approach C**). This approximation resulted in less uncertain MLH results in the mid-afternoon.

[Figure]

**Figure 2: An example of the effective MLH evolution for different approximation of Rn concentration in the residual layer. Approach C was used in the model. Spring average diurnal profile of $C_{Rn}$ in Ljubljana was used for comparison. $E_{Rn}$ was fixed and does not match the values finally used in the manuscript.**

The effective MLH calculated using the three different approaches mentioned above, were compared for the selected day of lidar measurements at AJ location (Fig. 3). When compared with lidar backscattered signal, daily evolution of the MLH by Approach B (constant Rn concentration in residual layer) gives too steep increase of MLH from the morning towards noon.

[Figure]

**Figure 3: Comparison of effective MLH calculated using the Approach A (black dotted line), B (violet dashed line) and C (black solid line, min and max). Due to low Rn concentration in the residual layer from the previous day, the Approach A and C gives similar results in this case.**

After calibrating the radon model for the best estimate of the effective $E_{Rn}$, lower $E_{Rn}$ was obtained with Approach C than with the Approach A, especially for the LJ location. $E_{Rn}$ estimates were in the range from 200 – 300 Bq m$^{-2}$h$^{-1}$ and 200 to 350 Bq m$^{-2}$h$^{-1}$ for LJ and AJ location, respectively. Considering that the scaling factor for 1 m a.g.l. in LJ can be around 2 or even higher (compared to 1.87 obtained for 2m a.g.l. measurements presented by Griffiths et al. (2013)), the $E_{Rn-0}$ would be in the 100 – 150 Bq m$^{-2}$h$^{-1}$ range, which is similar to the range reported by Karstens et al. (2015). In the case of AJ, the expected range of $E_{Rn-0}$ would be from 130 – 230 Bq m$^{-2}$h$^{-1}$ (in the case of approximate scaling factor of 1.5). In order to evaluate the bias introduced by the choice of $E_{Rn}$, results of BC emission rates were calculated for the 100 Bq m$^{-2}$h$^{-1}$ range of $E_{Rn}$ values (min – max) indicated in the Table 2.

Table 2 of the main text is updated accordingly.

**Table 2: Selected Rn exhalation rates ($E_{Rn}$) and the range (min – max) for each month for both measurement locations.**

| Month | $E_{Rn}$ (Bq m$^{-2}$ h$^{-1}$) | |
|---|---|---|
| | **Ljubljana** | **Vipava valley** |
| November 2016 | / | 200 (150 – 250) |
| December 2016 | / | 200 (150 – 250) |
| January 2017 | / | 300 (250 – 350)* |
| February 2017 |  200 (150 – 250) |  300 (250 – 350)* |
| March 2017 |  250 (200 – 300) |  300 (250 – 350) |
| April 2017 |  250 (200 – 300)* |  300 (250 – 350)* |
| May 2017 |  300 (250 – 350) |  350 (300 – 400) |
| June 2017 |  300 (250 – 350) | / |

* High uncertainty

By using the corrected radon model (Approach C) and the best estimate of radon exhalation rate obtained using calibration described above, the results slightly differ from the first version of the manuscript. All the figures and tables with model results, as well as discussion were updated. The most important changes are presented below:

**Corrections in the manuscript, Page 12, lines 1 – 8**

where $C^+_{Rn_{i-1}}$ represents radon concentration remaining after decay in the residual layer from the previous afternoon. An approximation of linear decrease of $C^+_{Rn_{i-1}}$ with height was considered, reaching radon concentration of zero at the top of the previous day's residual layer.

When MLH reaches its full extent in the late afternoon, it can extend above the previous day's residual layer, thus incorporating Rn "free" air into the ML. ~~In these conditions $C_{Rn}$ reaches its lowest daily concentration, which can be similar or even lower than concentration from the previous day's residual layer ($C^+_{Rn}$). In such conditions, calculation following Eq. 11 becomes unstable with high uncertainty, leading to significant overestimation of the effective MLH. Since incorporation of residual layer into the Eq. 11 resulted in most cases in non-meaningful determination of MLH in the couple of hours preceding the afternoon transition to SNBL, residual layer Rn concentration was set to zero ($C^+_{Rn_{i-1}} = 0$).~~

**Determination of MLH based on BC concentration vertical profiles**

The uncertainty of MLH$_{BC}$ (MLH derived from BC vertical profiles) estimates reported in the Table S2 is actually not the uncertainty of the MLH, but a standard error of the fitting parameter. Uncertainty of MLH$_{BC}$ is estimated to be around 50 m and this will be explicitly stated in the section 2 (Supplement), as marked below.

**Corrections of the Supplement (Section 2)**

**2. BC vertical profile measurements by ultralight aircraft over Ljubljana**

Black carbon vertical profiles (Figure S3) were measured in the Ljubljana basin using an ultralight airplane (Aerospool Dynamic WT9; see GLWF, 2019). The air was sampled using an isokinetic inlet and a modified version of the Aethalometer AE33 with 1 second time resolution (Drinovec et al., 2015). The location of the inlet prevented self-pollution from the airplane exhaust and the inlet was designed to be iso-kinetic at the airplane airspeed. The inlet is a conical diffusor, mounted on the holder of the Pitot tube under the wing, and designed for airspeed 240 km/h. The plane followed the helical path between 400 m and 1100 m  a.s.l. (100 – 800 m a.g.l. ). The BC concentration was used as a parameter quantifying the influence of ground sources on the primary air pollution in the mixing layer and the mixing layer height was estimated from BC vertical profiles (MLH$_{BC}$). The measured data was fitted using a Boltzmann function:

$$y = \frac{A_1 - A_2}{1 + e^{(x - x_0)/dx}} + A_2 \,,$$

where $x_0$ represents the mixing layer height (MLH$_{BC}$). Comparison of MLH$_{BC}$ determined by plane measurements and Rn-model (MLH$_{Rn}$) are presented in Table S2. The uncertainty of derived MLH$_{BC}$ is estimated to be 50 m.
The lower and upper ranges of MLH$_{Rn}$ fall within the uncertainty range of MLH$_{BC}$.

**Table S2: Summary of MLH determined from BC vertical profile measured by plane (MLH$_{BC}$) and MLH determined by Rn box model (MLH$_{Rn}$) (data for the closest hour is reported).**

| Date & time (UTC) |  MLH$_{BC}$ (m a.s.l) / (m a.g.l.) |  MLH$_{Rn}$ (m a.g.l.) |
|---|---|---|
| 16/02/2017 15:03 | 710/ 410 |  370 (270 – 460) |
| 09/03/2017 7:40 | 480 / 180 |  200 (160 – 240) |
| 15/03/2017 7:10 | 460  / 160  |  100 (80 – 110) |
| 19/05/2019 5:10 | 490  / 190  |  140 (120 – 160) |

**Uncertainty of the mid-afternoon estimates of the radon mode**

We agree that it is necessary to point out the uncertainty of the model, especially during the mid-afternoon conditions of convective days. The FFT filter with the cut-off frequency of 0.25 h$^{-1}$ can introduce some uncertainty due to eliminating real variation of Rn concentration beside the instrumental noise. However, the instrumental noise causes very unstable model results, since shifting between the two modes of PBL evolution (growing and shrinking) is observed during the whole day. Since running window averaging affects the daily amplitude of radon signal, we decided to use the FFT filter, which can decrease the instrumental noise, especially in the mid-afternoon when $C_{Rn}$ reaches the lower limit of detection.

Due to the uncertainty of the mid-afternoon results of BC emission rates, the calculation of daily averages was corrected. Instead of excluding mid-afternoon results, which can affect especially traffic related BC emissions, the weighted average was applied, introducing lower weight (50%) for the mid-afternoon $E_{BC}$. This approach would still account for the afternoon emissions, but with lower impact to avoid positive bias of results.

An additional section was added to the end of the manuscript (Page 20), expanding the discussion of the model uncertainty estimation.

[revised manuscript text omitted]

**5. Scope of investigation**

After close examination of site characteristics, including the wind field and its effect on the radon concentration, we constrained the criteria for AJ location to average daily wind speed below 2 m/s (instead of 2.7 m/s), which excludes 40% of data with the strongest wind.

We have shown, that the radon concentration diurnal profile does not depend on the wind direction for wind speed up to 1.5 m/s, which is enough to obtain reliable results for days of stable atmospheric conditions. Most important, despite of the relative proximity of the Adriatic coast, the air flow from the SW is not commonly observed. For the dataset when valid conditions of the model are met, the strongest wind speeds were present in the mid-afternoon from the west, in direction of the Vipava valley, for which radon exhalation rate is expected to be spatially homogeneous.

Although the radon-box model has a higher uncertainty at the AJ location and the number of days, when conditions meet the criteria for the box-model are limited, we believe that the results of black carbon emission rates and the discussion regarding the uncertainty of the radon-box model approach are valuable and can serve as an important example for the future studies conducted in complex terrain.

---

## Author Comment (AC2) · 15 Sep 2020

**Answers to the Referee 2**

We thank the anonymous Referee 2 for positive review of the manuscript. The Referee 2 especially recommended to rewrite and shorten the Introduction and a part of Methodology section. To address the raised issues, we will answer the comments one by one and provide the changes made in the text. Due to major revision of the manuscript proposed by Referee 1, we will implement the changes to the new version of the manuscript.

1. **The introduction section is unusually long with full of irrelevant information. I would suggest the authors modify the introduction section and re-write. Instead of describing the general impacts and roles of BC in the atmosphere (which are widely available in the literature), focus your description on the existing emission estimates, their problems, and the need for the methods which have been described in the manuscript. Two pages would be more than sufficient for the introduction.**

Thank you for this comment. We agree that by rewriting the Introduction section we can draw reader's attention to the importance of the presented method. In this regard, we have shortened the:
   - introduction to black carbon (BC)
   - information on worldwide BC emission inventories
   - information about the existing approaches for mixing layer height (MLH) determination
   - introduction to radon characteristics

Based on the comments of Referee 1, the information of modelled values of radon exhalation rate were included in the Introduction section (Pages 2 – 5).
* * *

[revised manuscript text omitted]

**2.** **Section 2.1 is also full of unnecessary information. Page 6, line 13-31: please reduce the content. There is no need to describe the population, growth, and implementation of various plans by the municipality. Please merge the 'measurement locations' and 'geological setting' together.**

As proposed by the Referee 2, the paragraph "Measurement locations" and "Geological setting" were merged and the section was shortened. However, from the point of view of interpretation of obtained results, we think it is important to provide an information regarding population density and characteristics of BC sources for both locations. When implementing abatement measures, such as traffic restrictions, the emission rate is the quantity which quantitatively demonstrates the efficiency of the measures taken.

The section "2.1 Measurement locations" was changed as follows:
* * *
**2.1 Measurement locations**

[revised manuscript text omitted]

**4. Page 9, line 12: Please add the full form of FFT in the list of abbreviations in Table 1.**

Table 1 was updated with adding the FFT to the list of abbreviations.

| Acronym/Symbol | Definition | Units |
|---|---|---|
| $\lambda'_s$ | temporal decay constant | h$^{-1}$ |
| AAE | absorption Ångström exponent | |
| AAE$_{BB}$ | biomass burning related AAE | |
| AAE$_{TR}$ | traffic related AAE | |
| AJ | Ajdovščina location | |
| ARSO | Slovenian Environmental Agency | |
| $b_{abs}$ | absorption coefficient | Mm$^{-1}$ |
| BC | black carbon concentration | ng m$^{-3}$ |
| BC$_{BB}$ | biomass burning related black carbon concentration | ng m$^{-3}$ |
| BC$_{TR}$ | traffic related black carbon concentration | ng m$^{-3}$ |
| C | multiple scattering parameter | |
| C$_{Rn}$ | radon activity concentration | Bq m$^{-3}$ |
| C$_s$ | species concentration | |
| E$_{BC}$ | black carbon emission rate | µg m$^{-2}$ h$^{-1}$ |
| E$_{BB}$ | black carbon emission rate from biomass burning sources | µg m$^{-2}$ h$^{-1}$ |
| E$_{TR}$ | black carbon emission rate from traffic sources | µg m$^{-2}$ h$^{-1}$ |
| EMEP | The European Monitoring and Evaluation Programme | |
| E$_{Rn}$ | radon exhalation rate | Bq m$^{-2}$ h$^{-1}$ |
| E$_s$ | species emission rate | |
| FFT | Fast Fourier transform | |
| GDAS | Global Data Assimilation System | |
| GOL | Golovec Astronomical and Geophysical Observatory | |
| h | effective mixing layer height | m |
| LJ | Ljubljana location | |
| ML | mixing layer | |
| MLH | mixing layer height | m |
| NOAA-ARL | NOAA Air Resources Laboratory | |
| OT | Otlica Meteorological observatory | |
| PBL | planetary boundary layer | |
| PM | particulate matter | |
| SNBL | stable nocturnal boundary layer | |
| T | air temperature | °C |
| wd | wind direction | |
| ws | wind speed | m s$^{-1}$ |
| $\gamma_s$ | spatial decay constant | m$^{-1}$ |
| σ$_{air}$ | mass absorption cross section | m$^2$ g$^{-1}$ |

**5. Section 2.4: It is recommended to provide a scatterplot of modelled-MLH with GDAS also in supplementary file.**

Since the same comment was also given by Referee 1, we have improved the description of methodology used for MLH comparison and the results, including the scatterplot, were added to the Supplement. A more detailed reply is provided in the Answers to Referee 1.

6. **Page 14, line 23: Authors mention that the Average Radon activity concentration was similar at both measurement locations, i.e., 15±11 Bqm⁻³ and 14±10 Bqm⁻³. At the same time, the authors also mentioned that it was slightly lower in the spring 13±9 Bqm⁻³ and 12±8 Bqm⁻³. Consider the standard deviations in the data, I do not see any difference in the data. Authors should check whether these differences are statistically significant or not and add a line on it.**

We agree with the observation pointed out by the Referee 2. There is in fact no significant difference in the seasonal averages of radon concentration measured at both locations. The text was corrected accordingly.

Page 14, lines 23 to 28:

Average radon activity concentration derived from hourly measurements (Figure 3) was similar at both measurement locations, $15 \pm 11$ Bq m$^{-3}$ and $14 \pm 10$ Bq m$^{-3}$,  in winter _and $13 \pm 9$ Bq m$^{-3}$ and $12 \pm 8$ Bq m$^{-3}$ in spring, in_ Ljubljana and Vipava valley, respectively (Table 4). These values are above annual average outdoor radon concentration of 10 Bq m$^{-3}$ reported by UNSCEAR (2000) for the continental areas_. Due to limited atmospheric mixing, higher winter concentrations are usually observed. However, decrease of atmospheric $C_{Rn}$ by more efficient atmospheric mixing is compensated by increased radon exhalation rate in the warmer season._ .

7. **Page 15, line 14: Despite significantly higher……………25% higher in LJ than in AJ. What is meant by only 25% higher? And how population is a factor here? It looks highly ambiguous statement. The authors should consider removing it.**

The statement was intended to point out the difference between BC concentration at both locations, based on the assumed stronger BC emissions in the area of denser population (Ljubljana) and subsequent traffic density. Since the average BC concentrations are discussed further on in the same section, this statement was removed.

8. **Figure 1 should be modified significantly. The background map items are almost invisible.**

Thank you for this observation. Fig. 1 was modified for clear visibility of all important items.

[Figure]

**Figure 1: Map of Slovenia (a) with marked areas of measurement sites Ljubljana (LJ) and Vipava valley (VV). b) The city  of Ljubljana with urban background (ARSO) and hill (Golovec – GOL) measurement sites. c) Area of the Vipava valley with urban background (Ajdovščina – AJ) and hill (Otlica – OT) measurement sites (Source: Map data ©2018 GeoBasis-DE/BKG (©2009) Google and OpenStreetMap)**